# Parametric slat design study for thick base airfoils at high Reynolds numbers

Julia Steiner[1], Axelle Viré[2], Francesco Benetti[3], Nando Timmer[4], and Richard Dwight[5]

[1,2,3,4,5]Faculty of Aerospace Engineering, Kluyerweg 1, 2629 HS Delft, Netherlands

**Correspondence:** J. Steiner (j.steiner@tudelft.nl)

**Abstract.** Standard passive aerodynamic flow control devices such as vortex generators and gurney flaps have a working principle that is well understood. They increase the stall angle and the lift below stall and are mainly applied at the inboard part of wind turbine blades. However, the potential of applying a rigidly fixed leading edge slat element at inboard blade stations is less well understood but has received some attention in the past decade. This solution may offer advantages not only under steady conditions but also under unsteady inflow conditions such as yaw. This article aims at further clarifying what an optimal two-element configuration with a thick main element would look like, and what kind of performance characteristics can be expected from a purely aerodynamic point of view. To accomplish this an aerodynamic shape optimization procedure is used to derive optimal profile designs for different optimization boundary conditions including the optimization of both the slat and the main element. The performance of the optimized designs shows several positive characteristics as compared to single element airfoils, such as a high stall angle, high lift below stall, low roughness sensitivity and higher aerodynamic efficiency. Furthermore, the results highlight the benefits of an integral design procedure, where both slat and main element are optimized, over an auxiliary one. Nevertheless, the designs also have two caveats, namely a steep drop in lift post-stall and high positive pitching moments.

## 1 Introduction

Generally, for the inboard part of wind turbines blades, thick airfoils with a high maximum lift and ideally a low roughness sensitivity are preferred over thinner profiles with high aerodynamic efficiency. Further, the installation of vortex generators ahead of the separation line is the current standard in the wind turbine industry (Rooij and Timmer (2003)). This helps delay flow separation to higher angles of attack and compensate for insufficient blade twist. Other flow control devices such as leading edge slats (Zahle et al. (2012); Gaunaa et al. (2012); Schramm et al. (2016)), gurney flaps (Salcedo et al. (2006)) and gurney flaps in combination with vortex generators (Storms and Jang (1994)) have also been considered to enhance the blade performance in the inner third of the blade span.

Generally speaking, stall control methods are more effective than circulation control ones for the inboard blade regions since these blade sections often operate at high angles of attack. Furthermore, given the low contribution to the overall power production of the blades, cost-effective passive methods are more appropriate than active ones.

All of the previously mentioned flow control devices except the leading edge slat have a working principle that modifies the flow near the trailing edge to either increase the stall angle or the lift at the design angle of attack for (quasi-) steady inflow.

Vortex generators introduce a streamwise vortex in the boundary layer. This vortex enhances the mixing of the free-stream flow into the boundary layer and as a consequence makes the boundary layer more robust against adverse pressure gradients. When these small devices with a height of the order of the boundary layer thickness are placed on an airfoil in front of where the separation line would be for the uncontrolled case, the static stall angle is delayed at the cost of a small drag penalty for pre-stall angles of attack (Baldacchino et al. (2016)). Moreover, depending on the exact configuration of the device a very abrupt stall behavior may be observed (Mueller-Vahl et al. (2012)). Several attempts at optimizing the vortex generator shape and placement for static operation point can be found in the literature (Godard and Stanislas (2006); Mueller-Vahl et al. (2012); Fouatih et al. (2016)). Vortex generators are not limited to static stall control but have also been used for dynamic stall control (Pape et al. (2012); Heine et al. (2013); Joubert et al. (2013a, b); Choudhry et al. (2016)). For dynamic stall control, the devices have to be located close to the leading edge such that the formation of the dynamics stall vortex is suppressed or at least delayed to higher angles of attack.

Gurney flaps if applied on the pressure side work by increasing the streamline curvature at the trailing edge and result in an upward shift of the lift curve (Liebeck (1978)). A downward shift of the lift curve is observed if the device is placed on the suction side. Of course, they also introduce a drag penalty which can be managed by appropriately sizing them (Salcedo et al. (2006); Bach et al. (2014)). Due to their location close to the trailing edge gurney flaps have negligible dynamic stall control capabilities (Bach (2016)). Microtabs which are applied close but not directly at the trailing edge exhibit more or less the same characteristics (van Dam et al. (2007)).

By contrast, leading-edge slats are located near the leading edge and have a more complicated working principle than vortex generators and gurney flaps. As Smith (1975) points out the slat does not lead to a blowing type of boundary layer control, but rather a combination of five dominant effects:

- *Slat effect:* The circulation on the slat element leads to a reduction of the pressure peak on the main element.

- *Circulation effect:* The circulation around the main element induces an upward velocity component at the slat trailing edge. In order to fullfil the Kutta condition at the slat trailing edge, an increased circulation around the slat is necessary. Thus, the circulation of the slat in the vicinity of the main element is increased as compared to the free-standing slat element only.

- *Dumping effect:* This effect is closely related to the circulation effect. The circulation around the main element induces a low pressure region around the main element leading edge, and thus at the slat trailing edge located in its vicinity. As a consequence the high outflow velocity of the boundary layer of the slat relieves the adverse pressure gradient on the slat element. Hence, separation problems are further alleviated.

- *Off-the-surface pressure recovery:* At least partially, the pressure recovery of the slat wake takes place away from the wall. This type of pressure recovery is more efficient than one in contact with a wall.

– *Fresh-BL effect:* Two fresh boundary layers are formed on both the slat and the main element of the configuration, this increases the resistance of the boundary layer to strong pressure gradients.

As a result of this complex interaction between the two elements, a properly designed slat element leads to a larger static stall angle as compared to the main element alone. However, the slat also has the potential to lead to an increase in drag and a decrease of the lift over drag ratio as compared to the base profile, especially for lower angles of attack.

Smith (1975) also states that for optimal performance of the configuration, the slat trailing edge should be placed and oriented relative to the base element such as to avoid confluent boundary layers.

The higher stall angle makes slat elements interesting for application at the inboard part of large wind turbine blades for two reasons. First, the high lift configurations would allow for a reduction of the chord length and hence the standstill loads. Second, due to insufficient blade twist, the inboard sections often operate in the post-stall regime. Hence, profiles with higher stall angles have the potential to increase the energy yield of the turbine.

Additionally, experimental investigations into the effect of leading-edge slats under dynamic inflow conditions can be found
in literature. They found that a fixed leading edge slat can help ameliorate the effects of dynamic stall for thin profiles as relevant for rotorcraft applications (Carr and Mc Alister (1983); Carr et al. (2001)). The experimental investigation of a VR-7 airfoil with a slat optimized for steady-state operation showed lower peak lift and pitching moment values, as well as, a reduced hysteresis amplitude as compared to the base element only (Carr and Mc Alister (1983)). There are multiple effects at work here - the delayed stall angle and the location of the slat near the leading edge can help suppress the formation of the Dynamic
Stall Vortex.

Of course, this is also interesting for wind engineering applications, where the reduction of fatigue loading is important as turbines keep growing in size. In particular, when a wind turbine operates in yaw, the inboard sections often operate under dynamic stall conditions.

While most of the work on slat element design has been done on thin profiles relevant for aerospace purposes, in the past 10
80 years a few publications tried to assess the potential of such a configuration for thick wind turbine airfoils.

Pechlivanoglou et al. (2010) measured lift and drag on a DU97-W-300 base element equipped with a slat element that was 12 % of the chord length of the base element. The slat element in question is the NACA-22 slat, first named by Weick and Noyes (1933), but designed and first tested by Weick and Wenziger (1933). As compared to the base element only, a stall delay of $\Delta_{\alpha_{\text{stall}}} \approx 9°$ and an increase in maximum lift coefficient of $\Delta_{c_L} \approx 1.0$ was observed at a Reynolds number of $\text{Re} = 1.3 \times 10^6$.
Zahle et al. (2012) designed and tested a slat element for a 40 % thick flatback airfoil. The main airfoil was a scaled version of the FFA-W3-360 profile. The slat element chord length was 30 % of the chord length of the main element. For the resulting profile, CFD results predicted a stall delay of $\Delta_{\alpha_{\text{stall}}} \approx 16°$ and an increase in maximum lift coefficient of $\Delta_{c_L} \approx 2.5$ along with a decreased roughness sensitivity. Moreover, beyond an angle of attack of $\alpha \approx 4°$ higher lift over drag ratios as compared to the main element were predicted. However, according to CFD results, this was also accompanied by a very steep lift drop post-stall
of $\Delta_{c_L} \approx 2.0$ which was however less abrupt in the wind tunnel measurements. Overall, despite the shortcomings of RANS turbulence modeling, the trends observed in the CFD computations agreed with the measurements. Further, the measurements showed strong wall interference effects at high angles of attack rendering the measurement results unreliable.

Later on, Gaunaa et al. (2012) extended the previously mentioned framework to design slat elements for an entire wind turbine blade between 10 % and 30 % blade span. The sectional slat elements were parametrized using Bezier splines, the relative positioning of the slat trailing edge and the slat angle. Subsequently, they assessed the performance of the full, rotating blade with the slat element. They used the DTU 10 MW reference turbine and simply retwisted the sections where there was overlap with the slat element. Their results showed an increase in the power and thrust coefficient of 1 % and 2 %, respectively. However, the increase in energy yield in the inboard part was accompanied by a decrease in energy yield on the outboard part. Nevertheless,the authors of the paper report an error in the geometry for this publication and hence the results may not be entirely reliable.

Manso Jaume and Wild (2016) designed an optimized slat element for the DU91-W2-250 base profile. Further, they also performed a combined shape optimization of both the slat and the base element for a base profile thickness of 25 % and an optimized slat chord length of roughly 25 % using a steady-state Reynolds-Averaged Navier-Stokes solver with the Spallart Allmaras turbulence model. The integrated design where both main and slat element were optimized showed better performance than the superimposed one in terms of aerodynamic efficiency and slightly less optimal performance in terms of maximum lift. Further, the integrated design was predicted to have a much more docile stall behavior than the superimposed one. Manso Jaume and Wild (2016) used a simple parameterization for the slat with six degrees of freedom four of which pertain to the relative positioning of the slat relative to the main element. The remaining two are the slat nose radius and angle. For the parametrization of the base element, three degrees of freedom were allowed, namely suction and pressure side length, as well as, vertical leading edge position.

Along the lines of what has already been done in literature, this study aims to assess the potential of two-element configurations with thick base elements of up to 50 % at Reynolds numbers close to real scale. The article will present a gradual buildup of the optimization complexity by first optimizing only the position of an existing slat element, then optimizing shape and position of an auxiliary slat element, and finally, the integral design approach where slat and main element are optimized simultaneously. The novelty in this article is twofold. First, a detailed analysis of the influence of the optimization boundary conditions on the optimal design is carried out. Second, the results of an integral design procedure are shown for thick main elements up to 50 % using a variable spline discretization for both the slat and the main element contrary to the simpler parametrization used by Manso Jaume and Wild (2016). Both of these aspects have not been discussed in literature before.

## 2   Optimization methodology

In a review paper on shape optimization for Aerospace applications, Skinner (2018) suggests that the typical design space for these applications show many locally optimal configurations. Hence, global optimization algorithms such as Genetic algorithms or Particle Swarm algorithms seem better suited for these types of problems if a close to optimal configuration is not known a priori. However, global optimizers can be quite inefficient in terms of the number of objective evaluations. This holds true especially towards the end of the optimization, once the approximate location of the local optima is identified. In this stage

of the optimization procedure, gradient-based optimizers would be more efficient than non-gradient based ones. Hence, as an alternative a hybrid optimizer can be used. Another option would be to use a surrogate model with a global optimizer.

For the sake of simplicity, in this publication, only a global optimizer in the form of a genetic algorithm is used in combination with a relatively cheap fluid model. Then, for the performance assessment post-optimization a higher fidelity fluid model is used. The different aspects of the optimization framework are summarized in the following:

– **Shape parametrization:** For the parametrization of both the slat and the base airfoil, Bezier spline curves are used. Two splines are used to represent the suction, as well as the pressure side of the profile. Additional constraints are imposed to ensure C0 and C1 continuity between the two sections. The leading edge of the main element location was fixed to the origin of the coordinate system $(0,0)$. A virtual trailing edge point is set at the point $(0,1)$. The trailing edge thickness is then assumed to be perpendicular to the x-axis of the coordinate system and the virtual trailing edge is set to lie in the 135 middle of the trailing edge line. Hence, the chord line always aligns with the x-axis. While, in principle, the trailing edge thickness can be part of the optimization procedure, for this article the trailing edge thickness is fixed.

For a derivative-free optimization algorithm, the number of spline points should be kept as small as possible to reduce the computational effort. Fitting existing airfoil shapes using six spline points for the suction side and the pressure side gave good results. Due to the geometric constraints at the leading and trailing edge, this leads to thirteen degrees of freedom 140 per element. An example of a spline point distribution and the corresponding profile shape is shown in Figure 1.

For the representation of the slat position relative to the base element, four degrees of freedom are used. These are: slat chord length $c_{\text{slat}}$, slat chord line rotation $\beta_{\text{slat}}$ relative to the coordinates system of the base element, gap width $h_{\text{slat}}$ between the slat trailing edge and the main element, and the surface distance $s_{\text{slat}}$ along the base element. The slat coordinate system and scaling are visualized in Figure 1.

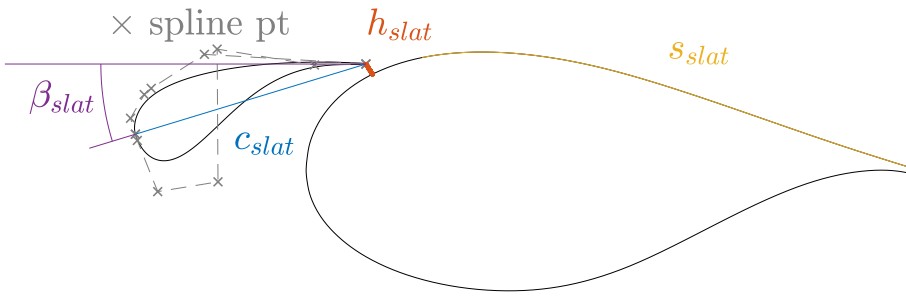

**Figure 1.** Visualization of the shape paramterization and the slat coordinate system paramterization (adapted from Gaunaa et al. (2012)).

– **Fluid models:** Drela's Integral Boundary Layer (IBL) code MSES and the computational fluid dynamics (CFD) code OpenFOAM-plus (also abbreviated as OF) are used interchangeably in the optimization framework. The specific settings

for the two models will be laid out on a case by case basis in the results section. The mesh for the CFD model is generated automatically. Because of the relatively complicated geometry, a hybrid mesh with structured blocks to resolve the boundary layer around the element is used. With the commercial mesh generator Pointwise, a combination of the hyperbolic extrusion and advancing front algorithms gave reasonable mesh quality.

– **Objective formulation and constraint handling** The optimization objectives $\text{obj}_{\text{lift}}$ and $\text{obj}_{\text{glide}}$ are formulated as a weighted sum of the performance under clean and rough condition. The performance under clean conditions respresents the case where natural transition occurs on both elements. Rough conditions assume that the transition is very close to the stagnation point on both elements, depending on the fluid model this is implemented a bit differently and explained in more detail in section 3.1. The performance is evaluated in terms of the normalized lift coefficients $\frac{c_{\text{L}}}{c_{\text{L,ref}}}$, as well as, as normalized glide ratios $\frac{(c_{\text{L}}/c_{\text{D}})}{(c_{\text{L}}/c_{\text{D}})_{\text{ref}}}$ for three different angles of attack $\alpha_j$. Hence, a multi-point, multi-objective formulation is employed as written out in Equation 1. The specific objective formulations are given in Equations 5 and 6 where $w_{\text{x}}$ are weighting terms. Additionally, a penalty formulation is used for the constraint handling. Constraints are imposed for the local airfoil thickness, the maximum thickness and the (previously mentioned) constraints related to the surface paramterization.

$$\textit{Minimize } \text{obj}_{\text{lift}}\left(\mathbf{x}\right),\ \text{obj}_{\text{glide}}\left(\mathbf{x}\right) \tag{1}$$

$$\textit{subject to } \text{con}_{\text{leq}}\left(\mathbf{x}\right) \geq 0 \tag{2}$$

$$\text{con}_{\text{eq}}\left(\mathbf{x}\right) = 0 \tag{3}$$

$$\mathbf{x}^{(\text{L})} \leq \mathbf{x} \leq \mathbf{x}^{(\text{U})} \tag{4}$$

$$\text{obj}_{\text{lift}} = \sum_{\alpha_j} \sum_{\text{clean/rough}} w_{\alpha_j} \cdot w_{\text{clean/rough}} \frac{c_{\text{L}}}{c_{\text{L,ref}}}(\alpha_j) \tag{5}$$

$$\text{obj}_{\text{glide}} = \sum_{\alpha_j} \sum_{\text{clean/rough}} w_{\alpha_j} \cdot w_{\text{clean/rough}} \cdot \frac{(c_{\text{L}}/c_{\text{D}})}{(c_{\text{L}}/c_{\text{D}})_{\text{ref}}}(\alpha_j) \tag{6}$$

– **Optimization framework:** The Python optimization toolbox Platypus is used which focuses on multiobjective evolutionary algorithms such as NSGAII, NSGAIII, and Particle Swarm optimization. The toolbox also allows for parallel objective evaluation using OpenMPI which was crucial for this project due to the high computational costs. For this study, only the NSGAII algorithm is used. Deb et al. (2002) describe the NSGAII algorithm implemented in the toolbox in detail.

## 3 Results

Subsection 3.1 will present a validation of the fluid models against experimental benchmark cases. The following three Sub-sections will highlight the results of three different design approaches: the optimization of the slat position for an existing slat design in Subsection 3.2, the auxiliary optimization of the slat element only in Subsection 3.3 and the integral optimization of the slat and the main element in Subsection 3.4. Finally, a comparison between the designs obtained from the last two design approaches will be presented in Subsection 3.5.

### 3.1 Fluid model validation

As already mentioned in the previous section, two different-fidelity fluid models are used for the optimization and the performance assessment post-optimization. The lower fidelity fluid model is Drela's commercial IBL code MSES and the higher fidelity model is the open-source CFD code OpenFOAM-plus. In this Subsection, the aim is to present a brief validation of both models on the benchmark case from Pechlivanoglou and Eisele (2014). Additionally, the CFD model will be validated against experimental results from the NHLP-90-L1T2 multi-element airfoil section (Moir (1993)).

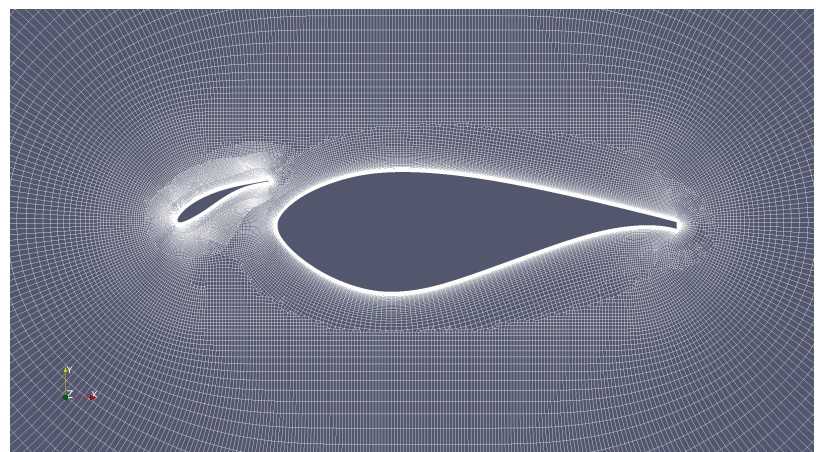

**Figure 2.** Benchmark configuration with DU97-W-300 base airfoil and a NACA-22 slat element.

The first setup consists of a DU97-W-300 base element and a 25 % chord length NACA-22 slat element (Pechlivanoglou and Eisele (2014)). An illustration of the profile and the CFD mesh are shown in Figure 2. The base element based Reynolds number is $\mathrm{Re} = 1.3 \times 10^6$. The turbulence intensity in the tunnel was around $\mathrm{TI} \approx 0.1\%$.

MSES solves the Euler equations on a discrete 2D grid, coupled with an integral boundary layer formulation. Transition is predicted by a semi-empirical $\exp^N$ envelope method. Some modifications have been made to this particular version of MSES to improve the stability of the code as detailed in van den Kieboom (2016). The mesh farfield distance was set to six chord lengths around the airfoil and a vorticity correction is used in the farfield. Otherwise, standard grid generation settings were used. To match the turbulence intensity in the wind tunnel the amplification factor was calculated with the correlation

$N_{crit} = -8.43 - 2.4\ln\left(\frac{TI}{100}\right) \approx 8.2$ from Drela (1995). For rough conditions, the boundary layer is tripped for both the slat and the main element at $x/C = 0.1\%$ and $x/C = 10\%$ on the suction and pressure side, respectively.

For the turbulence modeling in OpenFOAM a steady-state Reynolds-Averaged Navier-Stokes (RANS) model is employed. For the cases with rough conditions, the k-$\omega$ SST RANS turbulence model from Menter (1994) is used. Since this model is fully turbulent, transition does not occur and the boundary layer around the profile is fully turbulent. By contrast, for the cases with natural transition the correlation-based $\gamma$-$Re_\theta$ transition model from Langrty and Menter (2009) is used as an extension to the original k-$\omega$ SST model. Second-order schemes were used for the discretization of the momentum and continuity equations. For convergence reasons bounded first-order schemes were used for the turbulence model transport equations.

For the mesh generation, first cell height fulfills $y^+ < 1$ and the wall expansion ratio is chosen to be $\epsilon_{\text{wall-normal}} < 1.2$. As a result of a mesh convergence study the domain extent of the O-mesh was set to 400 chord lengths and around 300 points were used along the airfoil surface. The mesh is shown in Figure 2. To match the inflow conditions in the wind tunnel experiment, the inflow turbulence was set to TI $= 0.1\%$ and the eddy viscosity ratio was $\nu_t/\nu = 10$.

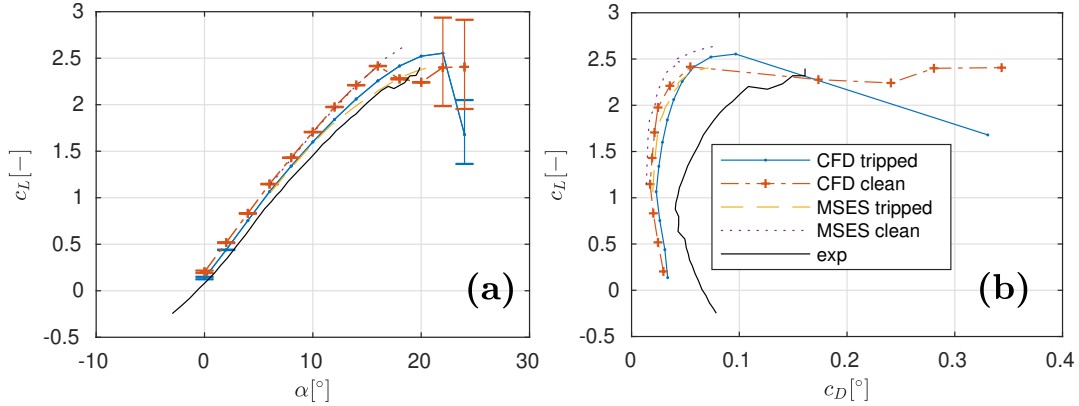

**Figure 3.** Fluid model validation against the benchmark case from Pechlivanoglou and Eisele (2014) for (**a**) the lift polar and (**b**) the drag polar.

The results from the two models and the wind tunnel measurements are shown in Figure 3. The profile in the wind tunnel was clean, but for the sake of comparison, the tripped configurations are shown in the Figures as well. Close agreement between the lift predictions of MSES and OpenFOAM are observed for both the clean and the rough case. MSES tends to underpredict drag as compared to CFD, but that is expected for IBL formulations. However, both models only yield satisfactory predictions for the lift coefficients and significantly underpredict drag coefficients as compared to the wind tunnel measurements. Because of the close agreement between the two models and because the two codes yielded very accurate results for the base element only, the author speculates that the discrepancy comes either from the experiment description or the experiment itself.

Yet, to investigate this discrepancy, the NHLP-90-L1T2 multi-element section from Moir (1993) was considered as a second benchmark case. The profile is pictured in Figure 4. The freestream Reynolds number, Mach number and turbulence intensity corresponding with the experiment are $Re = 3.52 \times 10^6$, Ma $= 0.197$ and TI $= 0.01\%$, where the Reynolds number is based on

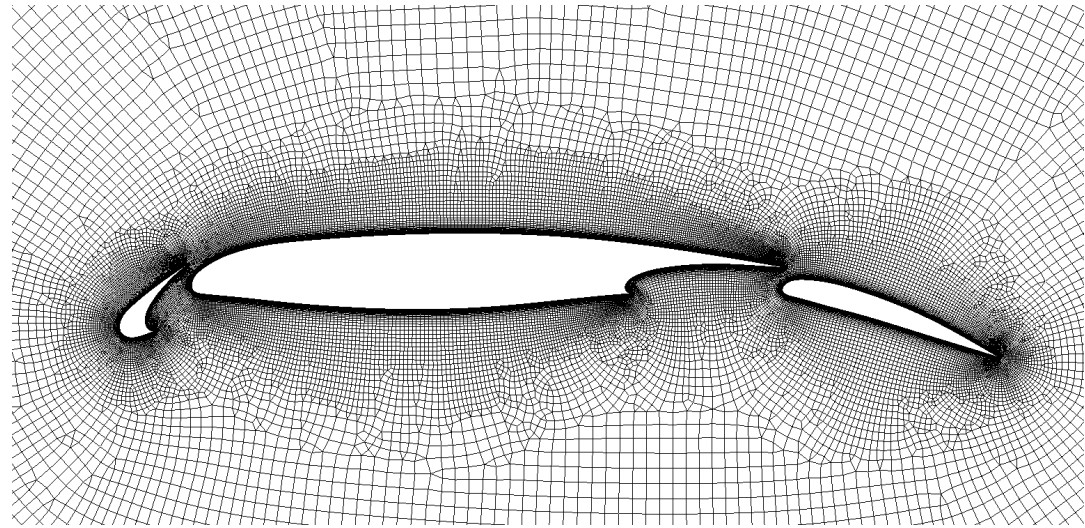

**Figure 4.** Coarsest considered mesh for the NHLP-90-L1T2 high lift configuration.

the length of the element when slat and flap are retracted. Although freestream Mach number is low, compressibility of the air
flow may affect the slat flow due to the high acceleration of the flow. Nevertheless, as will be seen later on, good agreement
between experimental and numerical results is obtained by an incompressible solver.

     The experiment is performed on a clean profile. Nevertheless, both the transitional and the fully turbulent simulations were
carried out. The results are shown in Figure 5. The results show an overprediction of the stall angle of about $6°$ and a slight
underprediction of the lift and drag below the stall angle. But, overall, the results are satisfactory and show that the employed
CFD model is capable of predicting both lift and drag with an acceptable error margin as long as only predominantly attached
flow is considered.

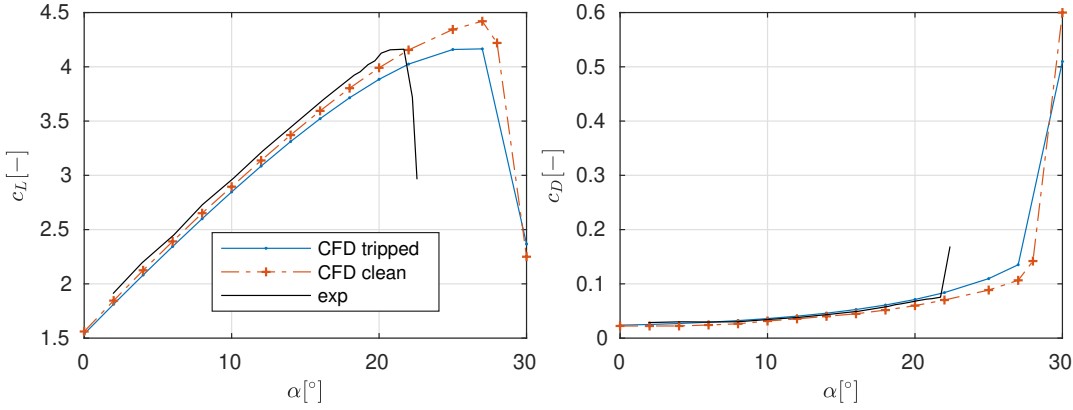

**Figure 5.** Fluid model validation against the benchmark case from Moir (1993) for (**a**) the lift polar and (**b**) the drag polar.

Unfortunately, MSES would not converge on this benchmark case due to the sharp corners of the profile. Nevertheless, correspondence between the two models was already established in the previous benchmark case. And since both MSES and OpenFOAM capture the same trends in results, it is concluded that for this particular case MSES can be used as a surrogate for the higher fidelity CFD model during the optimization procedure.

## 3.2 Slat dimensioning and optimization settings

The purpose of this Subsection is to establish sensible boundary conditions for the slat dimensions in the two following Subsections. Further, to reduce the computational cost of this step, a preliminary single-objective optimization of the slat position and dimension was carried out. The slat shape was fixed to correspond to the NACA-22 airfoil. As already mentioned, the slat position and dimension is parametrized using four degrees of freedom: the slat chord length $c_{\text{slat}}$, the slat angle $\beta_{\text{slat}}$, the distance between the slat and the main element $h_{\text{slat}}$ and the streamwise position of the slat element $s_{\text{slat}}$. However, to obtain sensible optimization results, either the slat chord length $s_{\text{slat}}$ or the gap width $h_{\text{slat}}$ needs to be constrained. Hence, two different fixed wall distances $h_{\text{slat}}/C = 4\%$ and $h_{\text{slat}}/C = 8\%$ were employed. The slat chord length was allowed to vary freely between $15\% \leq c_{\text{slat}}/C \leq 40\%$. The slat angle and chordwise position were not constrained.

The local chord-based Reynolds number for the NREL 5MW reference turbine at design conditions was calculated up to 40% span and was found to vary between roughly 9 and 10 million (Jonkman et al. (2009)). Hence, the design Reynolds number for the preliminary, and also the final cases, is chosen to be 10 million. Similarily, the freestream Mach number was also calculated at design conditions and it was below $0.1$ up to about 40 % span. Thus, an incompressible CFD solver as custom for wind energy applications was used. However, in MSES compressible effects were considered and hence the freestream Mach number was set to $0.1$ when using MSES.

The results are shown in Figure 6. The outcome shows that, independent of the prescribed slat height $h_{\text{slat}}$, the chord length of the optimal design converges to the upper bound, namely a slat chord length of 40 %. The performance coefficients of the two designs indicate that a larger gap between the slat and the main element also leads to higher lift and lift over drag ratios. However, this comes at the cost of a high positive pitching moment at the quarter-chord point of the main element. Since the slat would have to be attached to the base element, the quarter-chord point of the main element is a logical representative position for the calculation of the combined pitching moment of the profile.

Due to structural considerations, in the remainder, the slat height will be prescribed to be smaller than $h_{\text{slat}}/C < 4\%$. Further, two different chord lenghts will be investigated namely $c_{\text{slat}}/C = 30\%$ and $c_{\text{slat}}/C = 40\%$.

## 3.3 Auxiliary slat design

Now fixing the slat chord length and gap width, the results of the shape optimization of four different auxiliary slat elements are presented in the following. For the optimization, a two-objective formulation is used where the effects of the tradeoff between maximum lift and maximum aerodynamic efficiency are highlighted. Further, three different design angles of attack are chosen, namely $\alpha = 8°$, $\alpha = 12°$ and $\alpha = 20°$, where the second one is weighted the most. Also, the performance of the soiled profile is weighted much higher than the performance of the profile in clean conditions, since the first condition is more prevalent in

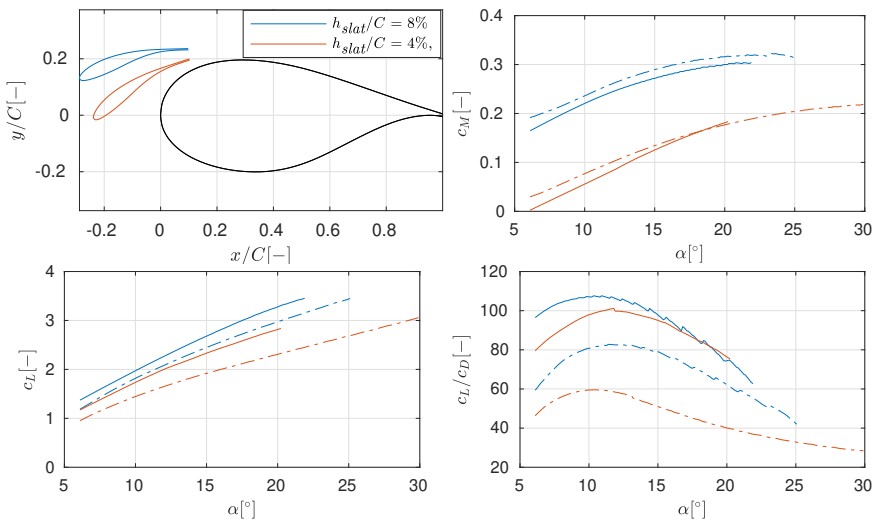

**Figure 6.** Results of slat position and dimension optimization as estimated with MSES for the clean ($-$) and the rough ($-.$) profile in terms of: **(a)** geometry, **(b)** lift, **(c)** drag and **(d)** pitching moment.

**Table 1.** General optimization settings (left) and optimization boundary conditions for the auxiliary slat design (right).

| Classification | Parameter | Value |
|---|---|---|
| *Flow regime* | Reynolds number | 10 million |
| | Mach number | 0.1 |
| | Amplification factor | 2 |
| *NSGAII settings* | Nr of design vars | 15 |
| | Mutation rate | $0.02 - 0.05$ |
| | Crossover rate | 0.8 |
| *Objective weighting* | $\alpha_j$ | $[8.0°, 13.0°, 20.0°]$ |
| | $w_{\alpha_j}$ | $[0.6, 1.0, 1.0]$ |
| | $w_{\text{clean}}, w_{\text{rough}}$ | $[0.2, 0.8]$ |

| Classification | Parameter | Value |
|---|---|---|
| *Conf. A* | Base profile | DU00-W2-401 (DU40) |
| | Slat chord $c_{slat}/C$ | 40% |
| | Gap width $h_{slat}/C$ | 4% |
| *Conf. B* | Base profile | DU00-W2-401 |
| | Slat chord $c_{slat}/C$ | 40% |
| | Gap width $h_{slat}/C$ | 2% |
| *Conf. C* | Base profile | DU00-W2-401 |
| | Slat chord $c_{slat}/C$ | 30% |
| | Gap width $h_{slat}/C$ | 4% |
| *Conf. D* | Base profile | FFA-W3-480 (FFA48) |
| | Slat chord $c_{slat}/C$ | 40% |
| | Gap width $h_{slat}/C$ | 4% |

reality. The general optimization settings as well as the boundary conditions for the different auxiliary slat designs are specified in table 1. The choice of boundary conditions documented there will be motivated in the following.

An optimized reference configuration referred to as configuration A with a slat chord length of $c_{slat} = 40\%$, a gap width of $h/C = 4\%$ and a base profile thickness of $t_{max}/C = 40\%$ is chosen. The base profile has the shape of the DU00-W2-401 profile. At this point, no structural constraints are considered. Hence, the only constraint that is imposed on the slat

geometry is that, at each chordwise position, the thickness is larger than the trailing edge. The trailing edge thickness is fixed to $h_{TE,slat}/C = 0.2\%$. As previously mentioned, in theory, either the slat chord length or the gap width need to be fixed to have a fully constrained optimization problem. However, the optimization results for both the preliminary and the actual designs tended to converge towards the upper bound for the gap width even if the bound was relatively large such as $h/C = 8\%$. Hence, the gap width was fixed in subsequent optimization runs to save computational cost.

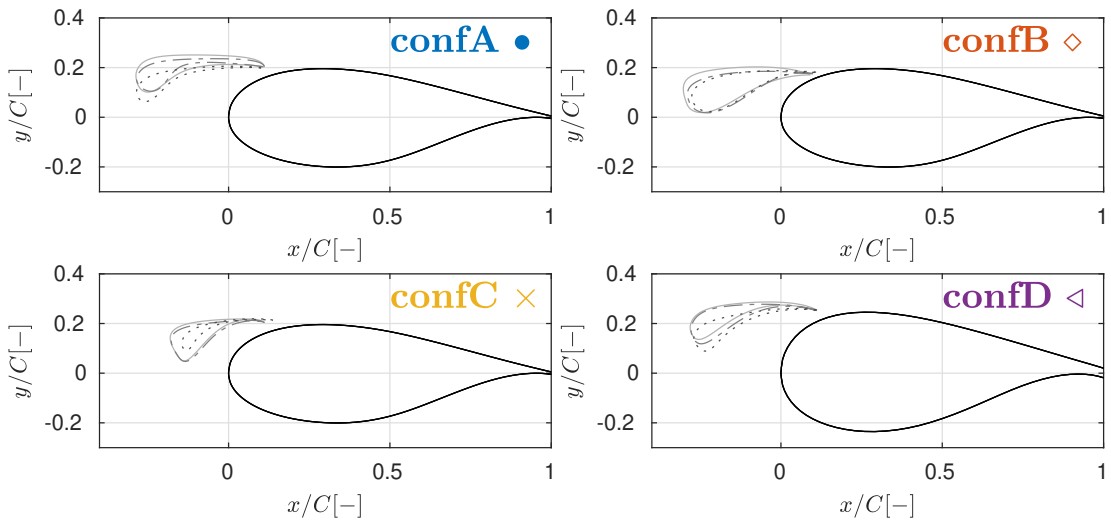

**Figure 7.** Designs obtained using the auxiliary design procedure optimized for maximum lift ($-$), maximum glide ratio (:) and a combination of both ($-$.), see figure 8 for clarification.

To assess the influence of the gap size between the slat and the main element, the slat relative chord length and the base profile thickness on the optimal slat design, the optimization procedure is also carried out for three modified design. For the second configuration named configuration B, the gap size is halved to $h/C = 2\%$ as compared to the reference design. For the third configuration referred to as configuration C, the slat chord length is reduced from $c_{slat}/C = 40\%$ to $c_{slat}/C = 30\%$. Lastly, for the fourth design named configuration D, a thicker base profile with a thickness of $t_{max,base}/C = 48\%$ is chosen. The

base profile is a scaled version of the FFA-W3-360 profile which was used in the The DTU 10MW Reference Wind Turbine Project Site.

    The optimal slat designs are pictured in Figure 7. Each time, three points on the Pareto front are shown. The three points correspond to the two extreme points and one point roughly in the middle of the Pareto front. For clarification see the example shown in figure 8. Figure 9 compares the different optimal slat designs in terms of slat angle, thickness, camber, and streamwise

position. The figure also already contains the results from the integral design procedure that will be presentend later on, so for now part of the results can be ignored.

    Given the limited sensitivity analysis shown in Figures 7 and 9, the following observations can be made:

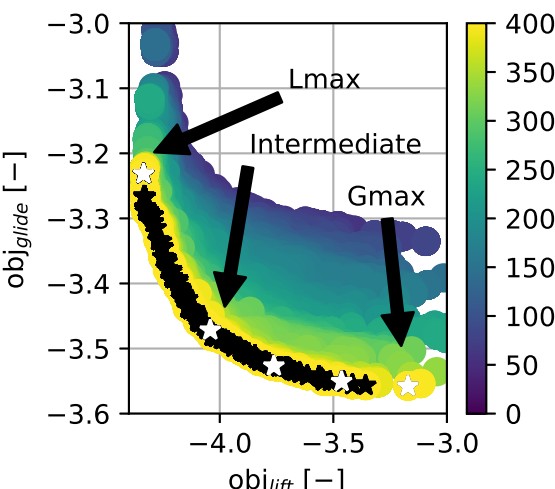

**Figure 8.** Pareto front example for the two-objective optimization with annotation of the two extreme and the intermediate objects. The color bars refers to the generation of the population.

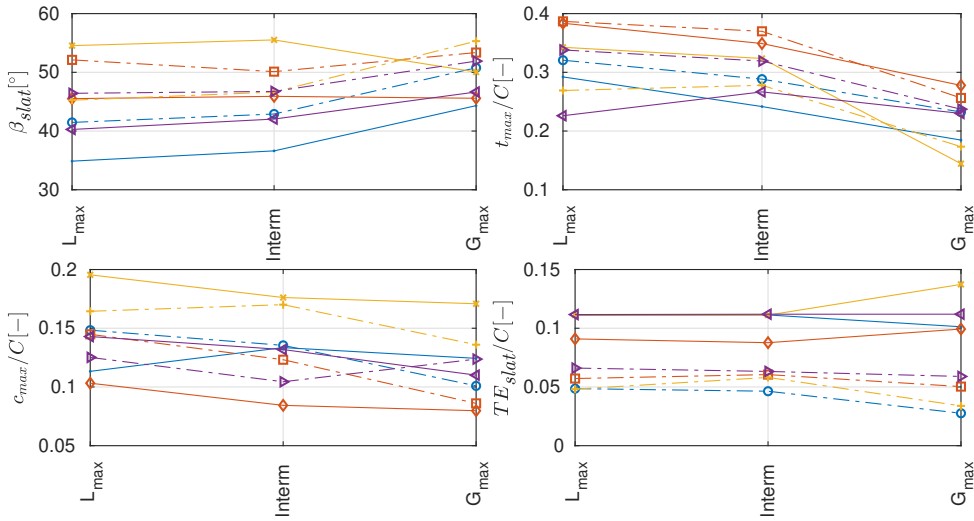

**Figure 9.** Slat characteristics from the auxiliary (−) and the integral (−.) design procedure in terms of **(a)** slat angle, **(b)** slat thickness, **(d)** slat camber, and **(d)** slat streamwise trailing edge location.

- *General trends:* The optimal slat design for all configurations and objective weightings are highly cambered, with cambers of the shown designs ranging between roughly 8% and 20%. This is a consequence of the previously mentioned dumping effect. Due to this, attached flow around the slat can be maintained even for a highly cambered profile. Moreover, the chordwise positioning of the slat is largely insensitive to the imposed design choices. The optimal streamwise

position of the slat relative to the base element is mostly determined by the streamwise location of the suction peak on the base element. Since all the base profiles are classical thick wind turbine airfoils, the location of the suction peak is very similar. Namely, the profiles have a relatively blunt leading edge that is designed to have the transition point close to the airfoil nose. Hence, the optimal streamwise slat position is around $x_{\text{TE,slat}}/C \approx 10\%$ for all the investigated constellations. As compared to more traditional high lift configurations for aerospace applications, the streamwise location of the slat element is further aft. This is, because usually for aerospace applications, thinner profiles which have the suction peak further forward are used.

– *Comparison with literature:* A comparison with the auxiliary slat optimization done by Schramm et al. (2016) and Manso Jaume and Wild (2016) for a 25 % thick base airfoil reveals similar optimal designs. Namely, the obtained slat designs have a large camber, the optimial slat streamwise position aligns with the location of the suction peak on the main element, a stall angle close to 20 degrees and a maximum lift increase of at least 100 %.

– *High lift versus high glide performance:* With exception of the design with the reduced chord length, the trade-off between high lift and high glide performance leads to smaller differences in the optimal designs than the change of the optimization boundary conditions. Except for one outlier, thinner and less cambered designs correlate with higher aerodynamic efficiency of the whole profile.

– *Influence of the gap width:* The reduction of the gap width between the slat and the main element leads to more downward turned, significantly thicker and less cambered optimal slat designs as compared to the baseline designs. This quite marked trend was also observed in other preliminary, unpublished designs (not shown here) with base airfoil thickness ranging between 25 % and 40 %. Aerodynamic theory indicates that reducing the gap width while avoiding confluent boundary layers leads to an increase in the coupling between the slat and the main element: the slat and the circulation effect are expected to get stronger whereas the dumping effect may be a bit weakened. However, the optimized slat for the lower gap width is less aggressive and the configuration produces less lift, has lower glide ratios and stalls roughly at the same angle of attack. Nevertheless, despite this somewhat counterintuitive result, the previously mentioned publications (Schramm et al. (2016), Manso Jaume and Wild (2016) and Pechlivanoglou et al. (2010)) use gap width of the same order of magnitude ranging between about $2.5\%$ and $6\%$.

– *Influence of the slat chord length:* The curtailment of the slat chord length leads to much higher cambered slat designs as compared to the baseline profiles. The higher camber indicates that a strong positive coupling exists between the slat and the main element since such a highly cambered profile would separate at fairly low angles of attack if used alone. The design optimized for maximum lift does not seem realistic and is solely exploiting a weakness of the fluid model. This will be further explained later in the article.

– *Influence of the base profile thickness:* The change in the base profile thickness introduces smaller design deviations from the baseline case as compared to the chord and gap width reduction. This goes back to the argument that the strongest design driver for the slat element is the location of the suction peak on the main element.

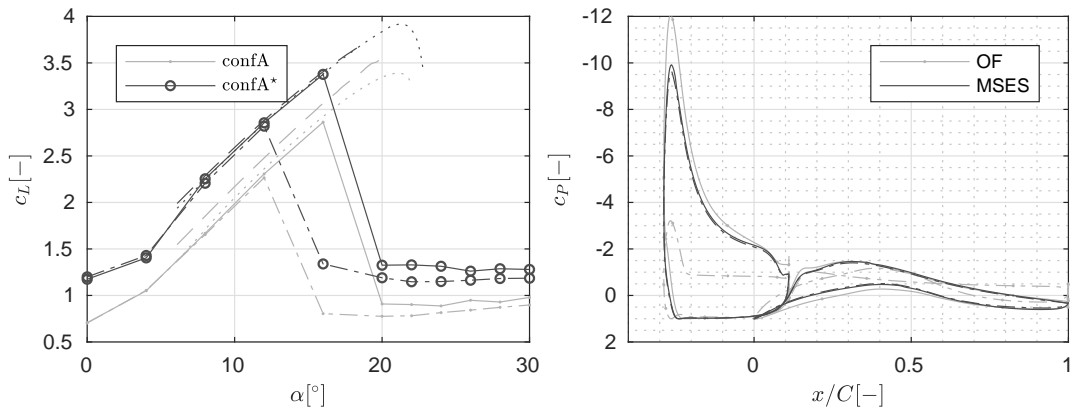

**Figure 10.** Performance assessment of configuration A cases optimized for maximum lift using either the auxiliary (caseA) or the integral design (caseA$^\star$) procedure. Shown in the figure are **(a)** the lift polar and **(b)** the pressure distribution at $\alpha = 16°$ as obtained using OpenFOAM ($-$ clean, $-$. rough) and MSES ($--$ clean, $\cdot\cdot$ tripped).

Moving on to a more detailed performance assessment of the optimal designs, the performance coefficients the maximum lift and the maximum glide ratio designs as estimated by CFD are shown in Figures 10 and 11, respectively.

Figure 10 shows a comparison between the lift predictions from MSES and OpenFOAM for the baseline configuration alongside the pressure distribution at $\alpha = 16°$. As was previously alluded to, the optimizer seems to exploit weaknesses in the fluid model for the high lift designs. Due to the setup of the optimizer, the flow model has to converge up to $\alpha = 20°$.

However, MSES tends to overpredict the stall angle by at least $\Delta_\alpha \approx 5°$ as compared to OpenFOAM. Further, as was already visible in the benchmark cases, the employed turbulence models for CFD do already overpredict the stall angle as compared to measurements. For the rough profile at $\alpha = 16°$, MSES predicts fully attached flow on both elements whereas OpenFOAM seems to indicate large separation zones on both elements. Hence, the profiles optimized for maximum lift at the design angles actually perform worse in terms of lift as compared to the ones optimized for maximum glide ratio at the design angles. Thus,

going forward only the designs optimized for maximum glide ratio will be considered. Due to the shortcomings of MSES in modeling the stall onset, it does not seem to be a suitable tool to be used in maximum lift design optimization. For the sake of completeness, the lift and drag polars for the high lift designs are plotted in Appendix A in Figure A1.

The performance of the profiles optimized for maximum glide ratio in Figure 11 shows some common characteristics despite the different optimization boundary conditions. For the rough profiles, the stall angle lies close to $\alpha \approx 20°$ and maximum lift

coefficients lie above $c_L > 3$. Beyond stall the lift coefficients steeply drop to $c_L \approx 1$. The roughness sensitivity is strongly reduced as compared to the base profile only, but this could also be a consequence of the low weighting of the clean performance in the objective formulation. Beyond $\alpha \approx 8°$, the glide ratio is larger than for the base profile only and the glide ratio shows a much less pronounced peak in the glide ratio distribution. The combined pitching moment calculated at the quarter chord point of the main element is positive due to the forward location of the slat element. The force and momentum coefficients for the

main element only, shown in Figure 11, are obtained from CFD as well, even though measurements are available.

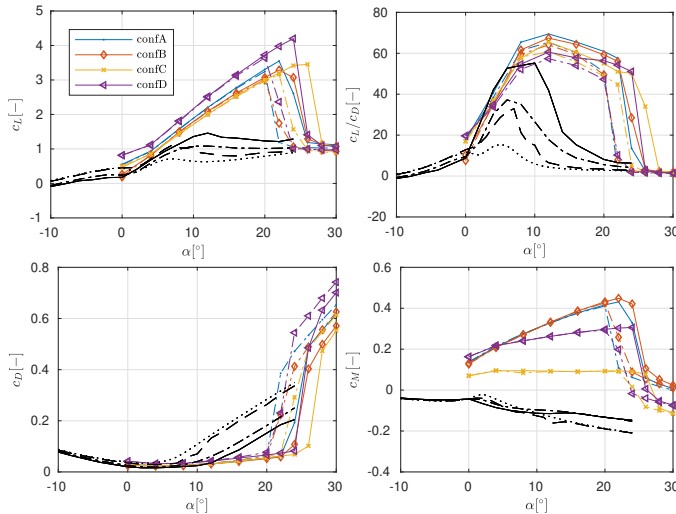

**Figure 11.** Lift (**a**), glide ratio (**b**), drag (**b**) and pitching moment (**d**) polar as obtained from CFD for the slat configurations optimized for maximum aerodynamic efficiency using the auxiliary design strategy. The coefficients for the clean (−) and tripped (−.) DU40 profile only, as well as, the clean (−−) and tripped (··) FFA48 base profile only are shown as well.

As compared to the baseline design, both a reduction in the gap between the main and the slat element, as well as a curtailment of the slat chord length, lead to lower lift and lift over drag values. As expected, an increase in the base element thickness leads to higher lift and drag coefficients. The largest spread in the performance coefficients between the different designs is seen in the pitching moment coefficient. The design with the curtailed chord length shows pitching moment coefficients of the

same magnitude as the main element only. The pitching moment of the other designs is between 2 and 4 times higher than that of the base element only.

### 3.4   Integral design of slat and main element

In this subsection, the results of an integral slat design procedure where both the shape of the main and the slat element are optimized simultaneously will be presented. Again, four different cases with the same boundary conditions as for the

design of an auxiliary slat design will be used. The general optimization settings and the optimization boundary conditions are summarized in table 2.

The fixed trailing edge thickness of the slat element remains unchanged. The trailing edge thickness of the main element is fixed to $h_{TE}/C = 1\%$. Additionally, to make the design more realistic additional thickness constraints are imposed on the main element. This is to ensure that there is enough space for the wing box. Analogous to Bak et al. (2014), a minimum local

thickness of $t(x/C)/t_{max} > 85\%$ was enforced at two chordwise stations, namely at $x_1/C = 15\%$ and at $x_2/C = 40\%$. This leaves space for a box length and height of at least $l_{box}/C = 25\%$ and $h_{box}/t_{max} = 85\%$, respectively. For the design of the

**Table 2.** General optimization settings (left) and optimization boundary conditions for the integral slat design (right).

| Classification | Parameter | Value |
|---|---|---|
| *Flow regime* | Reynolds number | 10 million |
| | Mach number | 0.1 |
| | Amplification factor | 2 |
| *NSGAII settings* | Nr of design vars | 28 |
| | Mutation rate | $0.02 - 0.05$ |
| | Crossover rate | 0.8 |
| *Objective weighting* | $\alpha_j$ | $[8.0°, 13.0°, 20.0°]$ |
| | $w_{\alpha_j}$ | $[0.6, 1.0, 1.0]$ |
| | $w_{\text{clean}}, w_{\text{rough}}$ | $[0.2, 0.8]$ |

| Classification | Parameter | Value |
|---|---|---|
| *Conf. A*★ | Base profile thickness $t_{max}/C$ | 40% |
| | Slat chord $c_{slat}/C$ | 40% |
| | Gap width $h_{slat}/C$ | 4% |
| *Conf. B*★ | Base profile thickness $t_{max}/C$ | 40% |
| | Slat chord $c_{slat}/C$ | 40% |
| | Gap width $h_{slat}/C$ | 2% |
| *Conf. C*★ | Base profile thickness $t_{max}/C$ | 40% |
| | Slat chord $c_{slat}/C$ | 30% |
| | Gap width $h_{slat}/C$ | 4% |
| *Conf. D*★ | Base profile thickness $t_{max}/C$ | 48% |
| | Slat chord $c_{slat}/C$ | 40% |
| | Gap width $h_{slat}/C$ | 4% |

slat element structural constraints are ignored at this point since they are considered less critical because both the bending and the torsional loads on the slat element are lower than the loads on the main element.

The resulting optimal designs are shown in Figure 12. Again three elements of the Pareto front are shown, the two most extreme ones and one roughly in the middle. At first sight, the designs look very similar to the ones from the previous Subsection where only the slat element is optimized. The optimal slat designs show the same sensitivity to the boundary conditions as for the design in the previous Subsection. However, their streamwise placement is much further forward than for the auxiliary slat design cases owing to the blunter leading edge of the main elements. The shape of the optimal main elements is much less sensitive to the aerodynamic efficiency of the design than the optimal shape and position of the slat element. However, this is at least partly a consequence of the structural constraints on the main element, because without the structural constraint especially the pressure side of the main element looked very different. Nevertheless, the main element profiles look very similar to the results obtained by Manso Jaume and Wild (2016). A more detailed comparison between the designs obtained with the auxiliary and the integral design procedure will be shown in the next Subsection.

The performance assessment of the optimal designs was again carried out using CFD. The result for the designs with the maximum glide ratio are shown in Figure 13. Since the designs optimized for maximum lift suffer the same shortcomings as for the previous optimization round, they are not discussed and only pictured in Figure A2 in Appendix A.

Again also the integral two-element designs optimized for maximum aerodynamic efficiency show some common characterictics. The stall angle lies around $\alpha \approx 25°$ and a steep drop in the lift coefficient to about $c_L = 1.5$ is predicted post-stall by CFD. The glide ratios lie above the ones for the main element only for $\alpha > 8°$. For some of the designs, this holds true even below that angle of attack. The glide ratio remains close to the maximum glide ratio for the angle of attack range of

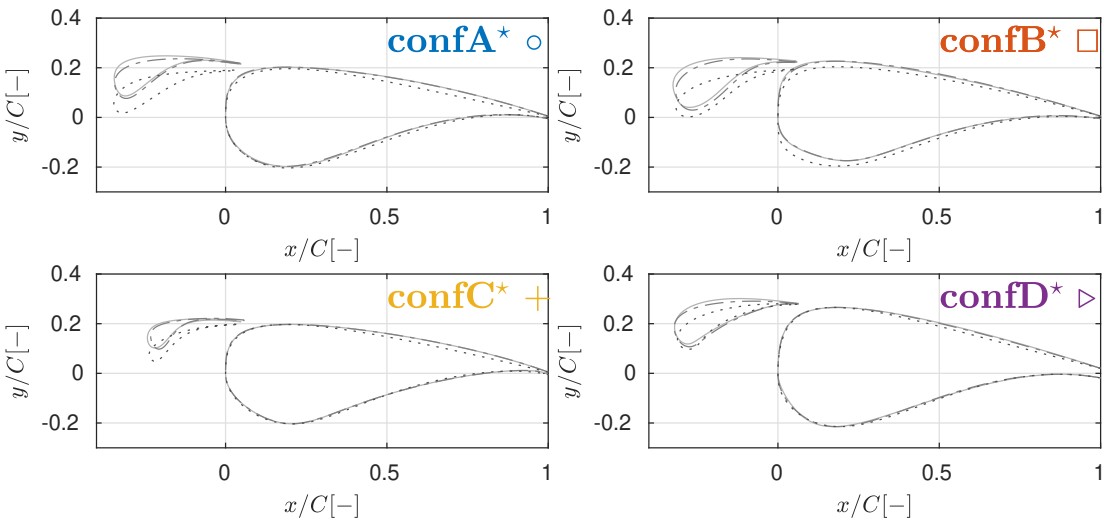

**Figure 12.** Designs obtained using the integral design procedure optimized for maximum lift ($-$), maximum glide ratio (:) and a combination of both ($-.$), see figure 8 for clarification.

roughly $8° < \alpha < 20°$. Further, a much lower roughness sensitivity than for the base element only is observed. Lastly, again the pitching moment coefficients show the largest dependence on the optimization boundary conditions.

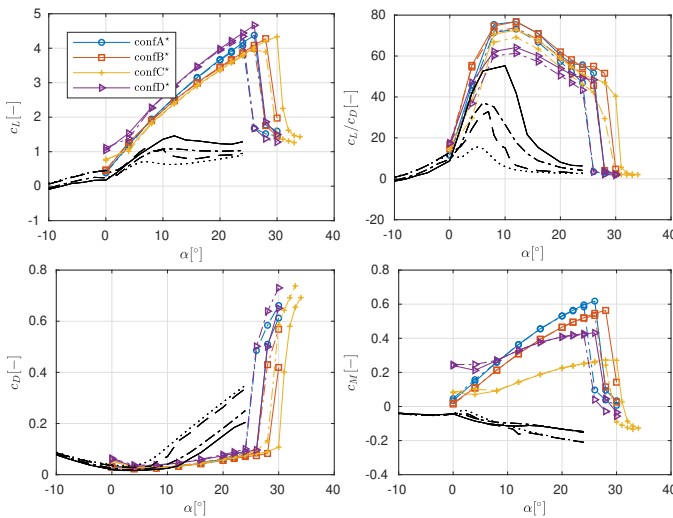

**Figure 13.** Lift **(a)**, glide ratio **(b)**, drag **(b)** and pitching moment **(d)** polar as obtained from CFD for the slat configurations optimized for maximum aerodynamic efficiency using the integral design strategy. The coefficients for the clean ($-$) and tripped ($-.$) DU40 profile only, as well as, the clean ($--$) and tripped ($··$) FFA48 base profile only are shown as well.

When comparing with the performance of the designs with the fixed main element from Figure 11, roughly the same sensitivity to the boundary conditions is observed and some performance gains are obtained through a more integral design approach independent of the boundary conditions. Namely, the stall angle increases by about $\Delta_{\alpha_{stall}} \approx 5°$. Further, at $\alpha = 20°$, an increase in lift coefficient and aerodyanmic efficiency of at least maximum $\Delta_{c_{Lmax}} \approx 0.5$ and $\Delta_{G_{max}} \approx 5$ is observed, respectively. On the flipside, due to the higher lift also the pitching moment increases, at $\alpha = 20°$, the pitching moment increases by about a factor of two for all the designs.

## 3.5 Comparison between designs with integrated and auxiliary slat design

In this Subsection, a more detailed comparison between the optimal designs from the auxiliary slat and the integral two-element procedure is presented.

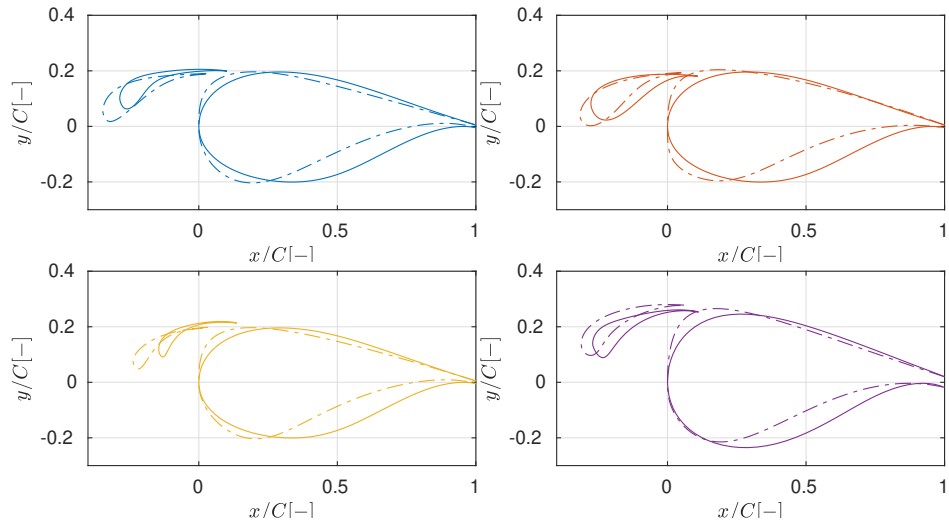

**Figure 14.** Comparison of profiles obtained using the auxiliary ($-$) and the integral ($-.$) design procedure.

Figure 14 shows the designs optimized for aerodynamic efficiency for the auxiliary and the integral runs. Three general trends are observed when comparing the integral with the auxiliary designs independent of the boundary conditions. First, the shape of the slat element excluding angle and position is not very dependent on the shape of the main element, but highly dependent on the optimization boundary conditions. Second, the optimal main elements show a much blunter leading edge and a lower profile thickness beyond $x/C \approx 35\%$ as compared to the standard wind energy airfoils used for the auxiliary slat design. The blunt leading edge leads to a forward shift of the suction peak, and hence, a forward shift of the optimal slat position as compared to the fixed main airfoils. The thinner shape of the main elements for the backward part of the profile may be a result of different wing box constraints, namely for the design of the wind turbine base airfoils possibly a larger box length was assumed. A more cambered main element leads to higher lift from the main element only which allows for less aggressive slat designs for the same total lift between main and slat element and higher stall angles.

In Figure 9, the slat angle, thickness, camber, and streamwise trailing edge location are shown both for the auxiliary and the integral design procedure. The two extreme elements and one element roughly from the middle of the Pareto front are shown. The range of the optimal slat thickness and camber is not strongly influenced by the choice of the design procedure. The slat angle and streamwise trailing edge location are more dependent on the choice of the design procedure. As already mentioned, the blunter leading edge of the main elements resulting from the integral design procedure leads to a more forward optimal slat trailing edge position. Additionally, the range in the optimal slat angle roughly halves when compared to the auxiliary design procedure. Likely, because a more forward position of the slat element reduces the influence of the shape of the base element on the optimal slat design.

In Figures 15, 17, 18 and 19 the pressure distributions for three angles of attack are shown for the designs optimized for maximum aerodynamic efficiency resulting from both the auxiliary and the integral design procedure. The pictured angles of attack are the two design angles $\alpha = 12°$ and $\alpha = 20°$, as well as an angle $\alpha = 28°$ that is either close to stall or post-stall, depending on the design.

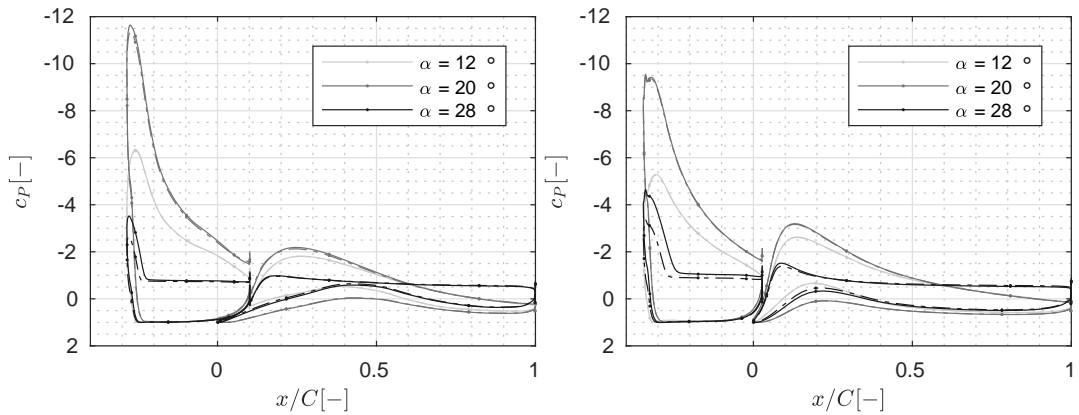

**Figure 15.** Pressure distribution from CFD for: the clean (-) and the rough (-.) profile at different angles of attack for **(a)** configuration A, and **(b)** configuration A$^{\star}$.

For the two baseline designs shown in Figure 15, the profile resulting from the auxiliary optimization procedure shows higher suction peaks on the slat element and lower suction peaks on the main elements for all the pictured angles of attack as compared to the profile resulting from the integral design procedure. This shows that the integral design procedure leads to profiles which better balance the inverse pressure gradients on the two elements. Hence, the flow around the resulting designs is expected to remain attached up to higher angles of attack.

Both profiles also show the same stalling behavior as visualized in figure 16 for the clean reference configuration obtained from the integral design procedure. Initially, simultaneously the flow on both the slat and the main element begins to stall from the trailing edge. As the angle of attack is further increased and the separation line moves towards the leading edge, the wake of the slat becomes wider as well. At some point, the slat wake extent grows so much that the low-pressure area in the wake leads to a relaxation of the pressure gradient along main element and hence the flow on the main element reattaches. This is in

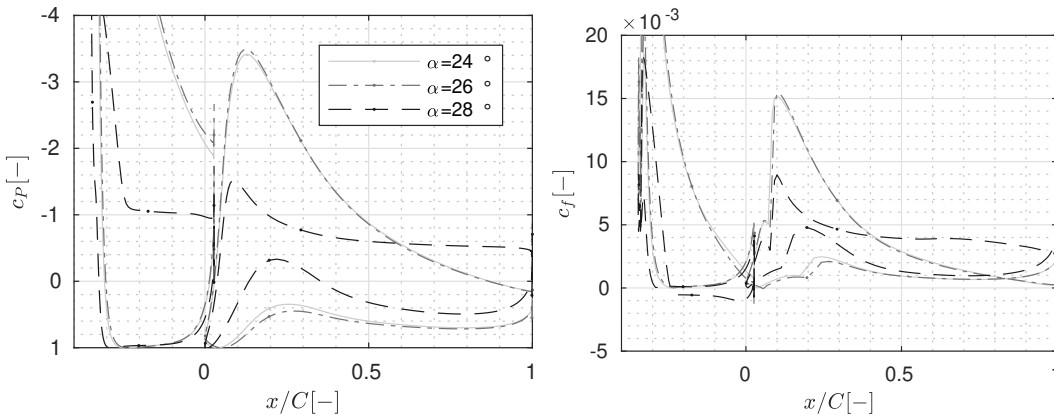

**Figure 16.** Stall progression for configuration C$^\star$ optimized for maximum glide ratio from CFD for the clean configuration: in terms of **(a)** pressure, and **(b)** skin friction coefficients.

accordance with the off-the-surface pressure recovery effect as identified by Smith (1975) and described in the introduction of
415 the paper. Finally, at some point, both elements are fully separated. In fact, this is observed for all the designs optimized for maximum aerodynamic efficiency in this article. See the Figures B1 and B2 in Appendix B for a visualization of this process for the configuration with the reduced chord length.

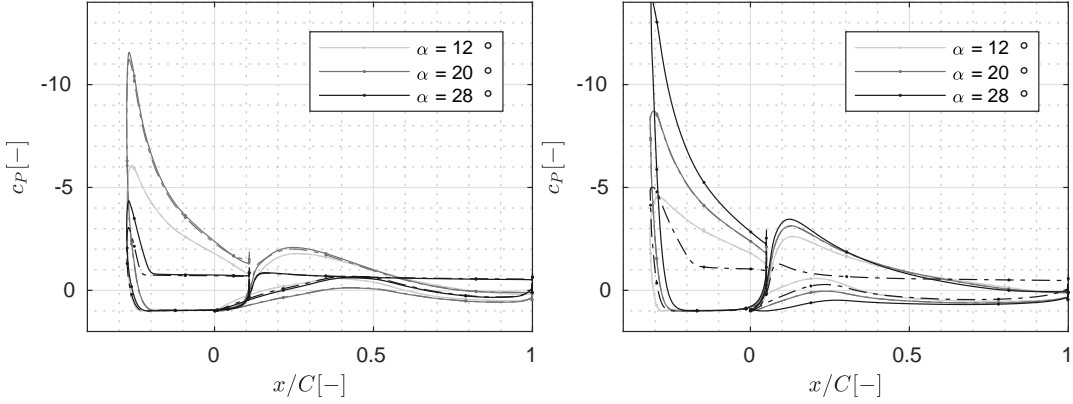

**Figure 17.** Pressure distribution from CFD for: the clean ($-$) and the rough ($-.$) profile at different angles of attack for **(a)** configuration B, and **(b)** configuration B$^\star$.

For the profiles with the smaller gap width shown in Figure 17, the pressure distributions look very similar to the baseline case. The main difference is that for these two configurations, the slat is turned into the flow a bit more. As a consequence, the
420 lift is a bit lower and the stall angle a bit higher as compared to the baseline design.

The pressure distributions for the profiles with the curtailed slat chord length are shown in Figure 18. The slat elements of these two profiles have the highest camber, the lowest thickness, and the highest slat angle when compared to designs

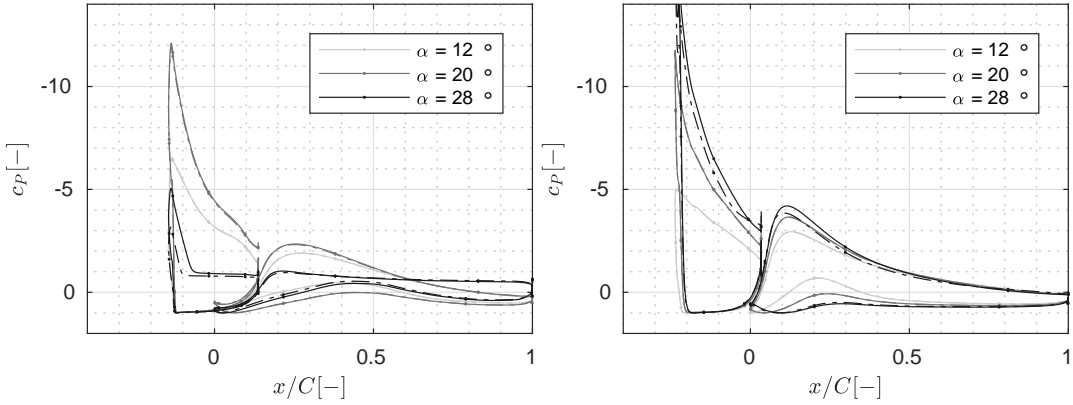

**Figure 18.** Pressure distribution from CFD for the clean (−) and the rough (−.) profile at different angles of attack for: **(a)** configuration C, and **(b)** configuration C$^\star$.

obtained with the same optimization procedure. The interaction between all these partially counter-acting effects leads to the highest suction peak on the main element and, as a consequence, the lowest pressure near the trailing edge on the slat element when compared to all the other profiles with the same main element thickness. This holds true despite the slightly lower lift coefficients when sized up with all the other designs. Consequently, as compared to the other designs, the main element is producing a larger share of the overall lift loading. This explains why the pitching moment coefficients are significantly lower for the designs with the reduced chord length. Further, they also show the highest stall angles within the two categories. Given that, as shown in Figures B1 and B2, the stalling mechanism is driven by the separation on the slat element, the reason for the higher stall angle could be the higher comparative circulation on the main element. This is because a higher circulation on the main element leads to a higher outflow velocity at the slat trailing edge, and hence, lower negative pressure gradients on the slat element.

Finally, Figure 19 shows the pressure distributions for the two profiles with the thicker main element. As visible in Figure 13, due to the higher base profile thickness, these two profiles have the highest lift coefficients and the lowest aerodynamic efficiency pretty much at all investigated angles of attack. However, beyond $\alpha > 8°$, the pitching moment coefficients are roughly in the midfield between the highest values from the thinner designs with the regular slat chord length and the lower values from the thinner designs with the curtailed slat chord length. The origin of this is twofold. First, the pressure distributions show a slightly lower ratio of the slat to the main element lift loading for the designs with the thicker main element as compared to the other ones. Second, for the integral design case, the optimal slat placement is a bit further aft as compared to the other designs.

Some of the designs show very high suction peak values for the slat which is an indication that locally compressibility effects may not be negligible despite the low freestream Mach number of $Ma_\infty \approx 0.1$. Calculating the local Mach number from the incompressible flow field for all the CFD cases shows that for the designs and angle of attack configuration where the lift coefficient is higher than 4 the Mach number locally approaches $0.45$. This is indeed very high and it is recommended

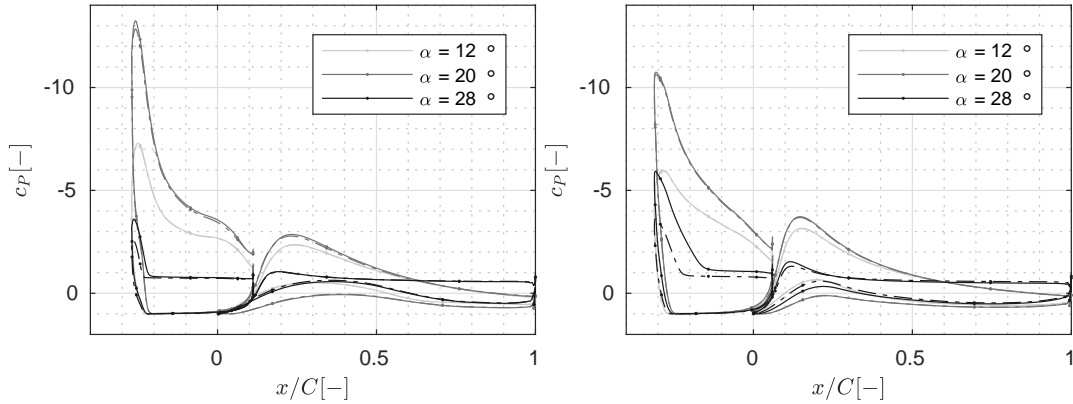

**Figure 19.** Pressure distribution from CFD for the clean ($-$) and the rough ($-.$) profile at different angles of attack for: **(a)** configuration D and **(b)** configuration D$^\star$.

that in future publications, compressibility effects should be considered if a design optimization for high lift is carried out. Nevertheless, given that the turbulence model of the CFD solver is expected to overpredict the stall angle, it is not certain that such high Mach numbers will actually be reached in real life.

## 4 Conclusions

This article comprises a parametric study on both auxiliary and integral slat design for thick main elements at a Reynolds number of 10 million. For both the auxiliary and the integral design procedure the influence of changes in the optimization boundary conditions on the optimal design is investigated. Initially, a baseline design with a slat chord length of 40 % and a gap width of 4 % of the chord length of the main element, as well as a 40 % thick main element is established. Subsequently, the influence of a 25 % reduction of the slat chord length, a 50 % decrease in the gap width and a 20 % increase in the main element thickness on the optimal design is documented.

All of the obtained profiles are predicted to have higher lift coefficients at almost all positive angles of attack, delayed stall, less roughness sensitivity, higher glide ratios above angles of attack of about 8 degrees, high positive pitching moments and a steep lift drop beyond stall. These effects were more pronounced for the designs obtained with the integral design procedure, as opposed to the ones obtained with the auxiliary design procedure. Hence, from a purely aerodynamics driven point of view, combined optimization of both elements offers additional advantages but also amplifies the caveats.

Comparison of the designs obtained with different boundary conditions gave rise to the following conclusions. While three out of the four investigated cases were carried out with a slat chord length of 40 %, a 30 % long slat element already offers many of the advantages of the two-element design without the caveat of a high positive pitching moment. While this comes at the expense of slightly lower lift coefficients, the stall angles are also higher. A reduction in the gap width did not offer any benefits, but only two gap widths were investigated. Possibly, the sensitivity to this parameter warrants further investigation.

Further, the increase in the stall angle and maximum glide ratio, as compared to the main element only, were more pronounced for the thicker main element. This indicates that this concept is more beneficial for thicker airfoils.

Summarizing, this article presented an analysis of the aerodynamic potential of slat elements for thick airfoils within the context of wind energy. Although this analysis highlights the benefits of a slat element on aerodynamic performance of wind turbine airfoils, multiple aspects still need to be further investigated. From an aerodynamic point of view, the next step would
be to investigate the influence of rotational effects. From a structural point of view, the next step would be to clarify how the significantly different aerodynamic performance would affect the structural design. The increase in the lift would allow a reduction of the chord length near the hub, which is beneficial for the standstill loads. However, it is still unclear how this would affect the scaling of the sectional bending and torsional stiffness, as well as, the blade mass. An increase in the box length may also be necessary. In particular, the positive pitching moment may make the occurrence of an aeroelastic instability,
such as divergence, more likely and the steep drop in lift post-stall may excite blade vibrations. Finally, logistics aspects such as the attachment of the slat to the main element also need further consideration.

*Code and data availability.* The open-source part of the codebase is available upon direct request with the correspondence author. Same goes for the data.

## Appendix A: Designs optimized for maximum lift

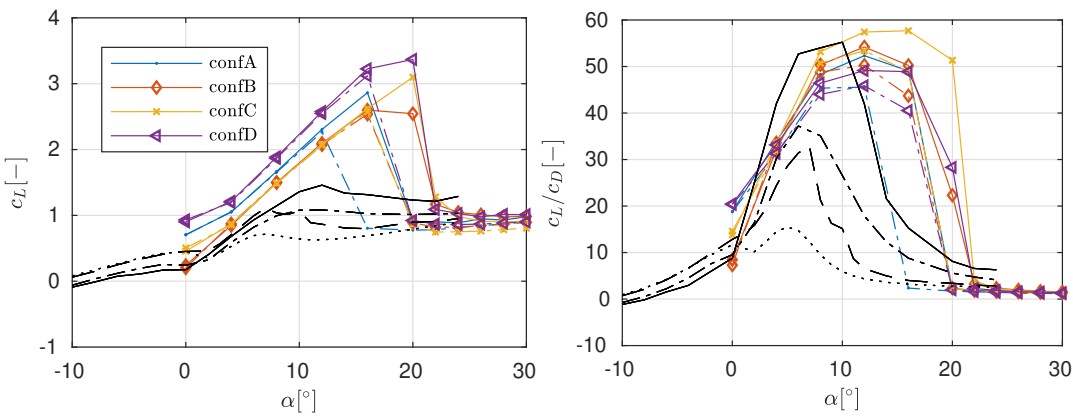

**Figure A1.** Lift **(a)** and glide ratio **(b)** polar from CFD for the slat configurations optimized for maximum lift using the auxiliary design strategy, the coefficients for the clean ($-$) and tripped ($-.$) DU40 profile only, as well as, the clean ($--$) and tripped ($\cdot\cdot$) FFA48 profile only are shown as well.

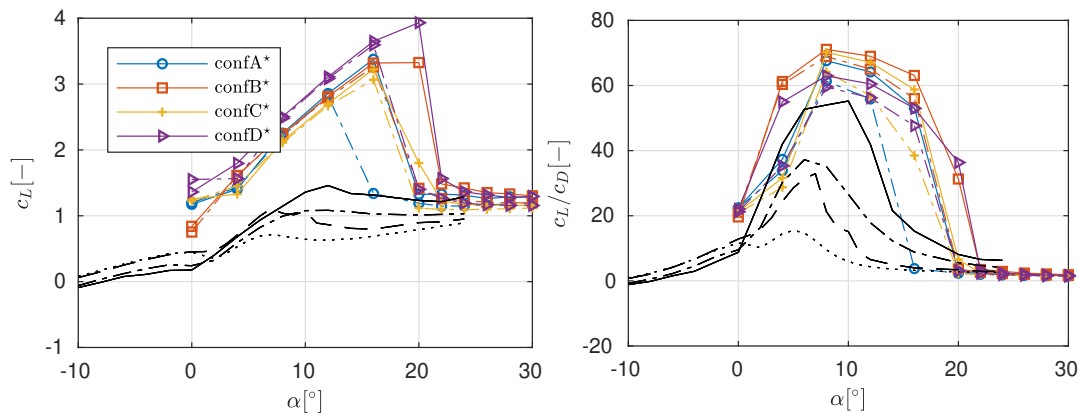

**Figure A2.** Lift **(a)** and glide ratio **(b)** polar from CFD for the slat configurations optimized for maximum lift using the integral design strategy, the coefficients for the clean ($-$) and tripped ($-.$) DU40 profile only, as well as, the clean ($--$) and tripped ($\cdot\cdot$) FFA48 profile only are shown as well.

**Appendix B: Stall mechanism on configuration C**

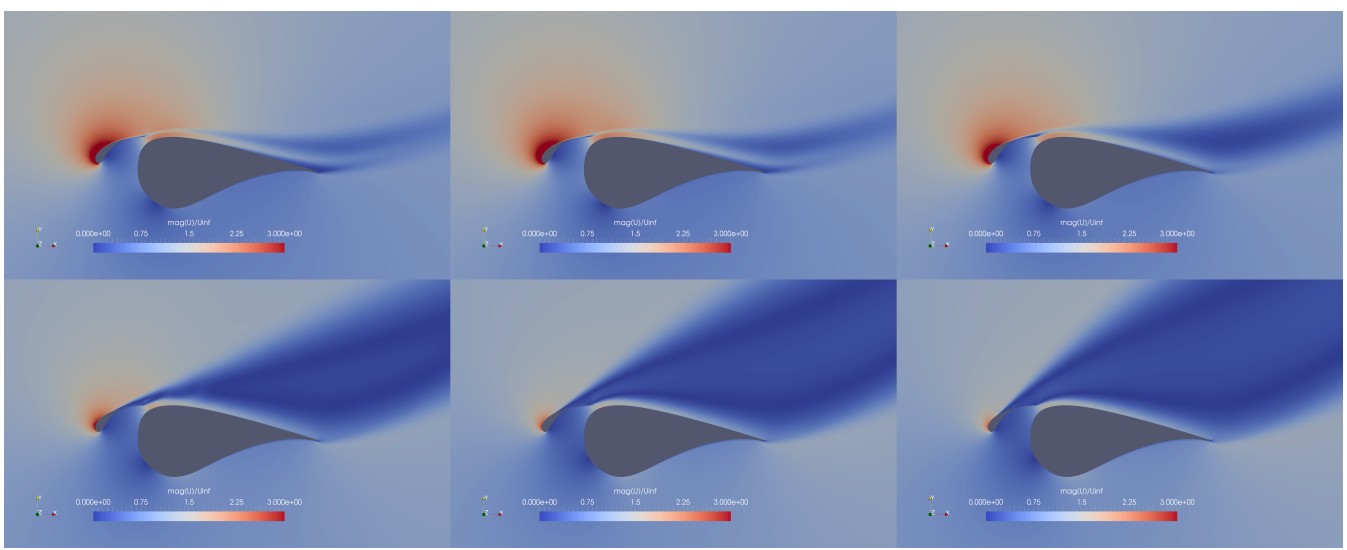

**Figure B1.** Flow fields around configuration C at the design Reynolds number and angles of attack ranging from $\alpha = 29°$ to $\alpha = 34°$ as predicted by CFD for the clean profile.

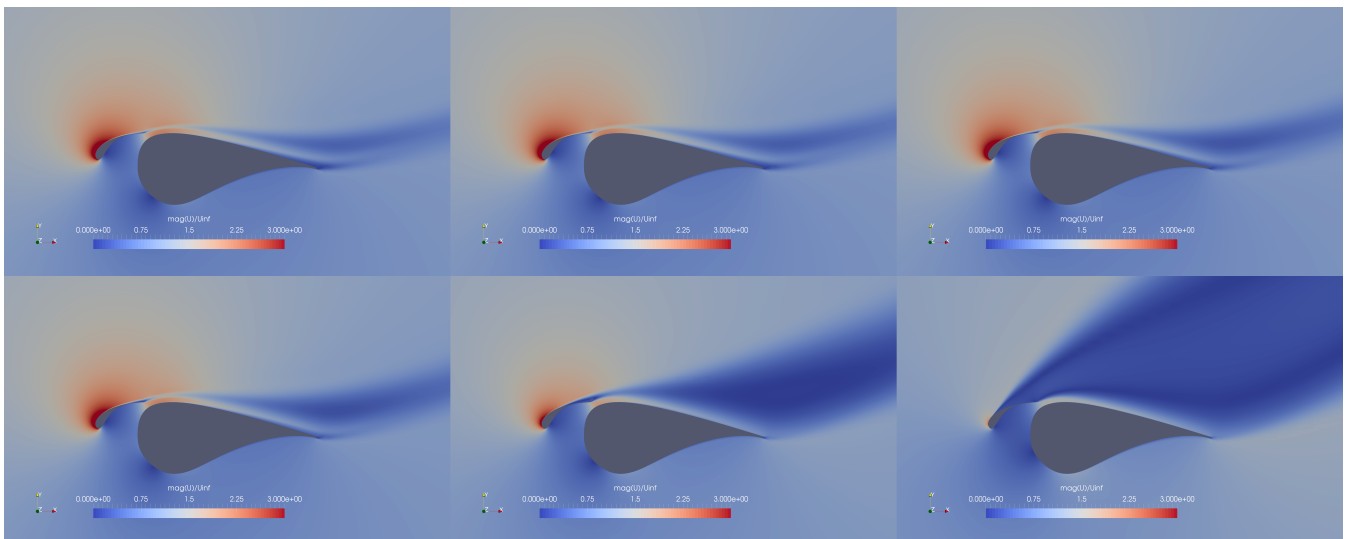

**Figure B2.** Flow fields around configuration C at the design Reynolds number and angles of attack ranging from $\alpha = 26°$ to $\alpha = 33°$ as predicted by CFD for the rough profile.

*Author contributions.* JS compiled the literature review, wrote the optimization codebase, carried out the design optimization and wrote the bulk of the paper. FB completed the validation of the CFD solver on the multi-element configuration. AV, NT and RD provided general input to the premise of the paper given their specific background, as well as proofreading of the manuscript.

*Competing interests.* The authors declare that they have no conflict of interest.

*Acknowledgements.* Steiner and Viré acknowledge support from the Rijksdienst voor Ondernemend Nederland (RVO) through the TSE Hernieuwbare Energie funding scheme (ABIBA project).

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
