# Peer review of "Parametric slat design study for thick base airfoils at high Reynolds numbers"

_Wind Energy Science, 2019_

## Referee Comment (RC1) · Anonymous Referee #1 · 6 Oct 2019

**General comments**

The work described in the paper is a very interesting design approach. The work as such is done in a good scientific manner. The wording is clear and good to read and understand.

Anyhow, the work needs major rework. Specific topics are highlighted in detail in the section with specific comments. In general, significant content that is moved to an appendix should be placed in the main text. Pressure distributions are shown but not discussed to a sufficient extent. Graphics have to be reworked and supported by descriptive captions and should be appropiately discussed in the text. Further on, the paper fails to compare the own work with previous experience (esp. the cited work done by Pechlivanoglou et al., Schramm et al., Manso Jaume and Wild. All of them already

addressed the slat design for wind turbine airfoils) to highlight common or contradictory results and thereby to identify specialties of thick baseline airfoils, which are highlighted to be one of the new contributions of the paper.

**Specific comments**

page 2, lines 27-43: The described methods for stall delay (vortex generators and Gurney flaps) are not part of the study. It should be checked if this information is of any benefit for the paper.

page 2, lines 47-48: The description of the Circlulation Effect is misleading. The first sentence is not describing the soure of the circulation increase. Instead, the circulation on the rear element induces an upward velocity component at the trailing edge of the preceding element. This has to be compensated by the forward element circulation to achieve the Kutta-condition at the trailing edge.

page 2, line 51: The description of the Dumping Effect does not describe the origin of the accelerated flow. It must be described that the high velocity at the forward element trailing edge is induced by the low pressure of the suction region at the leading edge of the downstream element.

page 3, line 57/58: It is a misunderstanding that the slat increases the lift coefficient at the same flow condition (angle of attack). This is usually not the case as long as the slat doesn't significantly increase the overall chord length of the airfoil system. This would be accomplished by a steeper gradient oft he lift curve vs.angle of attack. The lift created by the slat compensates the lift drop at the main element due to the reduction of the suction peak (Slat Effect). The part of the statement "an increase of the lift for all angles of attack and" should be deleted. Consequently, the text in the following paragraphs has to be adopted (delete "increased lift and" in line 63 as well as the sentence line 64/65).

page 3, line 79: The reference to the airfoil numbers is a correct citation of the referred work by Pechlivanoglou. Nevertheless, the notation of the TU Delft airfoil deviates from other known notations. Further on, the NACA 22 airfoil is at first not widely known and the notation suggests to be a mistake as there is no 2-digit NACA airfoil series as such. So it would be benefcial to refer to its origin (first named in Weick and Noyes, NACA TN 451, but designed and first tested by Weick and Wenzinger, NACA TR 407)

page 5, line 131: Is there an explanation why not a GC2 continuity is targeted. Especially in the leading edge region a curvature continous shape would provide smoother pressure distributions.

page 6, line 147: Please state what the authors assume to be a "reasonable" mesh resolution in more detail

page 6, line 151: It is unclear, where a local thickness is imposed.

page 6, lines 161-164: As the optimization framework and algorithms are not described in detail, proper reference and citation has to be given.

page 7, lines 173-174: provide citation of the reference to the codes used. For Open-FOAM make sure to refer also the code version and check-out date as open soure software tends to be changed very rapidly, but the reported results shall be reproducable.

page 7, line 176: Please refer to the airfoil correctly. The airfoil is called NHLP 90 L1T2 (see Woodward Lean, AGARD CP515) and it has been published by Moir as test case A2 for CFD validationdescribed in AGARD AR 303. The correction shall be propagated throughout the manuscript (e.g. page 8, line 201, caption of fig. 4 on page 9)

page 7, line 182: Are the six chord lengths sufficient in the view of the authors to eliminate effects on the boundary condition - or is there a vorticity correction at the farfield boundary employed?

page 7, line 191: It is stated that O-mesh topologies are applied although Pointwise is used. Please state, why not a C-mesh is used that would allow an improved capturing of the slat and main airfoil wakes.

page 8, line 200: The authors suspect the experiment to be the reason for the deviations, but it could be the missing resolution of the airfoil wakes, too.

page 8, line 203/204: This is a mistake. The Reynolds number in high-lift mulit-element airfoil cases is based on the "clean chord", which is the cruise airfoil with high-lift system retracted.

page 8, line 204/205: This is another - more common - mistake. Although the Mach number is relatively low, a look on the pressure peaks of this case unveils that the slat suction peak (although not shown here but reported in AGARD AR 303 or AGARD CP 515) gets into sonic speed conditions! Therefore, the choice of an incompressible solver for this airfoil is more than questionable.

page 8, line 208: The over prediction of the stall angle by $6°$ seems pretty large as the main motiviation of the work is based on the prediction of the stall delay by a slat which is mainly the shift in stall angle.

page 9, line 214: The conclusion that MSES can be used as a substitute for RANS CFD is weak and not supported. MSES is not able to capture confluent oundary layers at all. Due to the small gap and since the optimum slat position is very close to the position where the confluent boundary layer gets dominant (see Woodward and Lean, AGARD CP515, 1993) an optimization procedure neglecting this effect is liekly to predict gaps that are too small.

page 10, line 223: It is fully unclear why the most sensitive parameter for slat design - the gap - is fixed at the beginning. Additionally, the chosen values seem large. According to Woodward and Lean (1993) an optimum gap is strongly depending on the slat angle and can go down to 2-2.5% chord length. In the further (line 230 and following) the reason for the change in performance is most likely more related to the slat angle than the gap. It is consitent, that the optimal slat deflection angle is lower for the higher gap. At least concerning lift, it doesn't seem that a maximum lift coeffcient is clearly detected.

page 10, fig. 6: It is not consistient (and not expected by the reader) to show MSES results in these diagrams. Above it was mentioned, that the designs were optimized by MSES but the performance prediction for the evaluation is done with RANS. Especially, there is no clear max. lift coefficient prediction in the shown data.

page 11, line 258ff: an important description needed to understand the figures and the conclusions should not be placed in an appendix.

page 12, fig. 8: This figure is a collection of all optimization data. It is not very explanative as it overlays too much information. It contains already data (of the integral design) that hasn't yet even been introduced and is decribed much later. This figure should be divided for the different design methods and commented accordingly in the text.

page 12, lines 263-265: This would be a good option to highlight a common result with previous work (see General comments). This result is also in line with the results obtained by Manso Jaume and Wild for the superimposed slat optimization.

page 13, lines 278-281: The statement on the sensitivity of separation on the slat shape is not supported by theory. In constast, a cambered plate is less likely to separate at high angles of attack than a flat plate. Additionally, closing the

gap increases the slotted airfoil effects in both directions. In fact, as the slat is moved veritcally, the Slat Effect and the Circulation Effect are expected to get stronger. Only the Dumping effect is expected to be reduced due to the reduction of the suction peak due to the strengthened Slat Effect. In consequence, the slat load is increased (higher circulation and higher trailing edge pressure) resulting in a more cambered airfoil to be more suitable to achieve the circulation without separaton. To verify this, a comparison of the pressure distributions is needed. This conlclusion has therefore to be reworked.

page 13, line 281: Regarding the slat thickness, other preliminary designs are mentioned. It is unclear whether this is previous work (then to be cited) or own unpublished experience. Anyhow, this statement should be reworked to be reproducable by the reader.

page 13, fig. 9: The line legend of figure 9 introduces an undescribed configuration A* in addition (same for fig. 14, and in figs 15-17 configs B*, C*, D*). The meaning and origin is perfectly unclear. It can only be assumed from later reading that this configuration related to the integral design that is described much later (starting from page 15). The pressure distributions shown in the right hand side are not discussed at all in the text.

page 14, line 298: To be precise, none of the airofils is optimized for maximum lift coefficient. The airfoil optimization only targeted a high lift coeffciient at a high angle of attack (here AoA=20°). An airfoil stalling at 19° could have a higher maximum lift coefficient than one not stalled at 20°. To do a maximum lift coefficient optimization it is necessary to detect the naximum lift coefficient of an airfoil by varying the angle of attack.

page 14, line 310: Here it is stated that experimental data for the clean airfoil would be available for comparison. Such a comparison would have been an asset in section 3.1 regarding the validation of the methodology.

page 15, line 318: it should be highlighted that - in contrast to the integrated design work of Manso Jaume and Wild, where the suction side contour of the slat is the contour of the original main airfoil - the integral design here is not restricted by the clean airfoil shapein the same way. This underlined the originality of the present work and its relation to previous work.

page 15, line 334/335: It is mentioned that the main airfoil shape is a consequence of the structural constraints. But it is more expected that this is an implicit result of the main airfoil shape optimization. Due to the higher curvature, the suction peak is more locally concentrated (improving the Dumping Effect and stabilizing the slat flow) and the trailing edge position therefore placed close to the maximum curvature - which is now much furhter upstream. The intragrated design by Manso Jaume and Wild shows a similar main airfoil shape, and there, no structural constraint has been imposed.

page 17, lines 358-360: The discussion describes a "larger leading edge radius". It would be better to descirbe the leading edge as "more blunt". Further on, it is not a larger leading edge radius that shifts the suction peak to the front. A larger leading edge radius alone would reduce the suction peak but not move it. The present description is misleading as the curvature which is responsible for the interaction with the slat is higher (and the radius smaller) and imposes a rduced pressure at the slat trailing edge.

page 17, line 364/365: It is mentioned that the design angle doesn't account for lower side separaiton. This is correct, but anyhow, no clear indication of a lower side separation is seen for the designs with slat even at lower angles of attack.

page 17, lines 367/369: This is a very late explanation for a figure that had been placed on page 12. It is necessary to split-up fig. 8 and to place the related illustrations close to the dicsussion in the text.

page 17, lines 371: As already discussed "more rounded" suggests a smoother curvature distribution, while the opposite is the case for the integral designs.

page 18, lines 374/375: This conclusion is in contrast to a previous statement, where it was concluded that the optimal shape of the slat is not as sensitive to the main airfoil optimization and by this not as affected of the auxilary or integral design method (page 17, lines 356/357).

page 18, lines 383-385: Basically, to show the stalling behavior it is necessary to look at the pressure distribution just at stall onset. Best is a comparison with a very low AoA step before and after maximum lift. The used step of 8° is too large and - depending on the stall onset angle - the stall is developed over the entire configuration but not showing the onset (main wing or slat or both at the same time).

page 18, line 387: It should be discussed, whether the reattachment due to wake displacement is in accorcance with Snith's 4th effect (Off-Surface Pressure Recovery)

page 18, lines 387-389: As the stall onsiet is of primary interest, the discussion of the flow fields should not be placed into an appendix.

page 18, line 391/392: The last statement on this page is not directly supported by figure 15. To see this, it would be better to cross-plot the pressure distributions of both deigns at the same angle of attack on top of each other - and not side-by-side.

page 19, line 393: At least here at the end of all shown pressure distributions, it is necessary to conclude about the suitability of the incompressible solver. Although the flow speed is not mentioned (missing in the case description in Table A.1) the level of the pressure coefficient is less than -15 and imposes the need to check whether this assumption is still valid.

page 19, lines 395-397: The description reverses cause and effect. The high suction peak on the main airfoil is the reason for the low trailing edge pressure at the slat - remind Smith's effects.

page 21, line 428: To conclude on the importance of the gap it should have been used as a design parameter. The limited information on the gap variation (two values only) doesn't allowto draw such abbreviation general conclusion. From other airfoils in literature it is known that the gap is even the most sensitive parameter.

**Technical comments**

The Reynolds number is stated in the text in a form $Re = XXm$ with $m$ as an abbreviation to "million". As this could easily be mixed up with the unit "meter" a notation of the form $Re = xx \times 10^6$ would be recommended.

The reference to and citation of A.M.O. Smith's work is not proper. In the text, the citation "O. Smith (1975)" should read "Smith (1975)", e.g. on page 2 - line 45, page 3 - line 61. The reference (page 28 - line 507) should read "Smith, A.M.O.: ..."

Figures and captions must be improved. multi-figures are not fully described by captions. Labels (a) and so on are not clearly addressed tot the single figures. Line legends are often missing (especially fig. 8 and fig. 11).

Abbreviations and mathematical symbols shall be explained at their first occurence. Especially introducing abbreviations in figures only (e.g. "OF" in fig. 9) shall be avoided.

The word "subsection" is written with capital letter "S" at several locations. As this is not a name, a lower case "s" should be used (pages 9, 15 and 17).

The references list has to be reworked. DOI numbers appear twice, "https://doi.org" sometimes doubled. Journals are references by volume but not by number.

page 4, line 104/105: The citation "Manso and Jaume" should be replaced by "Manso Jaume and Wild" to be correct ("Manso Jaume" is the full last name of the first author - above in line 99 the citation is correct).

page 6, line 151: A line break before a sentence delimiter has to be avoided.

page 6, line 151: To be consistent in time "Constraints are imposed ..."

page 9, caption of fig. 4: "AGARD" is not a name but an abbreviation and should be written in capital letters

page 20, line 414: "compromises" should be replaced by "summarizes"

page 24, fig. B1: The color scale is unclear (which value plotted?).

page 25, fig. D1 and page 26, fig. D2 include angle of attack in plots. it is not clear whether to read the images left-to-right or top-bottom.

page 28, line 503: There is a mixup in the capital letters for the abbreviation of AGARD.

---

## Author Comment (AC1) · 23 Jan 2020

[1]JuliaSteiner

[Figure]

**Reply to the comment**

January 23, 2020

**General remarks**

Dear referee,

thank you for taking the time the review the paper in detail and for giving extensive feedback that helped us improve the draft. An insight from sombody with a more general aerospace based background was very useful for us.

Replies and adjustements to the specific and technical comments are outlined below. With respect to the general comments, most of them are addressed in the specific comments as well. Additionally, I added a couple more references & comparisons to the existing literature: see line 280ff/page 13, line 294ff/page 13, line 352ff/page 16 (expanded below).

- "Comparison with literature: A comparison with the auxiliary slat optimization done by Schramm et al. (2016) and Manso Jaume and Wild (2016) for a 25 % thick base airfoil reveals similar optimal designs. Namely, the obtained slat designs have a large camber, the optimial slat streamwise position aligns with the location of the suction peak on the main element, a stall angle close to 20 degrees and a maximum lift increase of at least 100 %."

- "Nevertheless, despite this somewhat counterintuitive result, the previously mentioned publications (Schramm et al. (2016), Manso Jaume and Wild (2016) and Pechlivanoglou et al. (2010)) use gap width of the same order of magnitude ranging between about 2.5% and 6%."

- "The shape of the optimal main elements is much less sensitive to the aerodynamic efficiency of the design than the optimal shape and position of the slat element. However,this is at least partly a consequence of the structural constraints on the main element, because without the structural constraint especially the pressure side of the main element looked very different. Nevertheless, the main element profiles look very similar to the results obtained by Manso Jaume and Wild (2016)."

**Reply to specific comments**
*page 2, lines 27-43: The described methods for stall delay (vortex generators and Gurney flaps) are not part of the study. It should be checked if this information is of any benefit for the paper.*
We consider the benefit of this information the option to highlight that the slat element has a different working principle and can be beneficial for different reasons. Furthermore, the beneficial effects of the two afformentioned elements are more widely known and hence to us it makes sense to mention them.

*page 2, lines 47-48: The description of the Circlulation Effect is misleading. The first sentence is not describing the soure of the circulation increase. Instead, the circulation on the rear element induces an upward velocity component at the trailing edge of the preceding element. This has to be compensated by the forward element circulation to achieve the Kutta-condition at the trailing edge.*
This was indeed misleading and has been reformulated to: "The circulation around the main element induces an upward velocity component at the slat trailing edge. In order

to fullfil the Kutta condition at the slat trailing edge, an increased circulation around the slat is necessary. Thus, the circulation of the slat in the vicinity of the main element is increased as compared to the free-standing slat element only."

*page 2, line 51:* *The description of the Dumping Effect does not describe the origin of the accelerated flow. It must be described that the high velocity at the forward element trailing edge is induced by the low pressure of the suction region at the leading edge of the downstream element.*
This was reformulated to: "This effect is closely related to the circulation effect. The circulation around the main element also leads to a low pressure region around the slat trailing edge. As a consequence the high outflow velocity of the boundary layer of the slat relieves the adverse pressure gradient on the slat element. Hence, separation problems are further alleviated."

*page 3, line 57/58:* *It is a misunderstanding that the slat increases the lift coefficient at the same flow condition (angle of attack). This is usually not the case as long as the slat doesn't significantly increase the overall chord length of the airfoil system. This would be accomplished by a steeper gradient oft he lift curve vs.angle of attack. The lift created by the slat compensates the lift drop at the main element due to the reduction of the suction peak (Slat Effect). The part of the statement "an increase of the lift for all angles of attack and" should be deleted. Consequently, the text in the following paragraphs has to be adopted (delete "increased lift and" in line 63 as well as the sentence line 64/65).*
The author agrees and this was removed. We do indeed see a lift increase for all angles of attack in our designs, but this seems to be a consequence of the higher apparent chord length.

*page 3, line 79:* *The reference to the airfoil numbers is a correct citation of the referred*

*work by Pechlivanoglou. Nevertheless, the notation of the TU Delft airfoil deviates from other known notations. Further on, the NACA 22 airfoil is at first not widely known and the notation suggests to be a mistake as there is no 2-digit NACA airfoil series as such. So it would be benefcial to refer to its origin (first named in Weick and Noyes, NACA TN 451, but designed and first tested by Weick and Wenzinger, NACA TR 407)*
The additional citation for the NACA-22 airfoil was added. The notation for the TU Delft airfoil was modified from DU97W300 to DU97-W-300 which is the notation used within TU Delft.

*page 5, line 131: Is there an explanation why not a GC2 continuity is targeted. Es- pecially in the leading edge region a curvature continous shape would provide smoother pressure distributions.*
This may be taken into consideration for further publications. Nevertheless, the pressure distribution shown in the paper do not show any irregularities possibly because a smoothing step is performed during the meshing routine. Further, the same shape parametrization was also used by Zahle et al. (2012) and Gaunaa et al. (2012).

*page 6, line 147: Please state what the authors assume to be a "reasonable" mesh resolution in more detail*
The mesh resolution is described later in the article on a case by case basis. For the evaluation of the new designs around 300 points were used along both slat and main element surface, y-plus was kept below 1 and the wall expansion ratios of below 1.2 were used for the structured part of the hybrid mesh.

*page 6, line 151: It is unclear, where a local thickness is imposed.*
The local thickness is defined as the thickness at a specific chordwise station of the profile. It is measured perpendicular to the chord line. The exact location of the local thickness constraints where applied are described later on in the template, namely at 15 % and 40 % chord length.

*page 6, lines 161-164:* *As the optimization framework and algorithms are not described in detail, proper reference and citation has to be given.*
Proper citation is given to the GitHub page of the optimization toolbox that is used. A link to the documentation of the toolbox was added in the bibliography. Additionally, a reference to a paper that explains the algorithm is added.

*page 7, lines 173-174:* *provide citation of the reference to the codes used. For Open-FOAM make sure to refer also the code version and check-out date as opensoure software tends to be changed very rapidly, but the reported results shall be reproducable.*
OpenFOAM-plus, version 1806 was used to obtain the results. For MSES a modified inhouse version was used. The references for OpenFOAM and MSES were already there, but in the fluid model validation section only. These were added in the framework description as well.

*page 7, line 176:* *Please refer to the airfoil correctly. The airfoil is called NHLP 90 L1T2 (see Woodward Lean, AGARD CP515) and it has been published by Moir as test case A2 for CFD validationdescribed in AGARD AR 303. The correction shall be propagated throughout the manuscript (e.g. page 8, line 201, caption of fig. 4 on page 9)*
This has been corrected in the entire draft.

*page 7, line 182:* *Are the six chord lengths sufficient in the view of the authors to eliminate effects on the boundary condition - or is there a vorticity correction at the farfield boundary employed?*
The sensitivity to the domain size was checked and it was bascially non-existant for the given domain size. Drela himself notes in the MSES manual that: "It must be stressed that the exact values of these grid parameters are not very important, since high-order vortex+doublet farfield representation makes the solution extremely insensitive to the

location ofthe outer grid boundaries." A note was added in the revised draft to include the vorticity correction at the farfield boundaries: "The mesh farfield distance was set to six chord lengths around the airfoil and a vorticity correction is used in the farfield."

**page 7, line 191:** *It is stated that O-mesh topologies are applied although Pointwise is used. Please state, why not a C-mesh is used that would allow an improved capturing of the slat and main airfoil wakes.*
This is a good remark and should be considered in further publications. However, in this paricular case with the automated meshing procedure using an O-mesh was more practical. Further, a thorough mesh sensitivity study was carried out.

**page 8, line 200:** *The authors suspect the experiment to be the reason for the deviations, but it could be the missing resolution of the airfoil wakes, too.*
At this point we cannot rule it out completely, but we have performed simulations on the main element only with an O-mesh at a Reynolds number of 2 million and we were able to match the experimental drag and lift coefficients below stall very well. So it seems unlikely that the O-mesh configuration is causing this large discrepancy.

**page 8, line 203/204:** *This is a mistake. The Reynolds number in high-lift mulit- element airfoil cases is based on the "clean chord", which is the cruise airfoil with high-lift system retracted.*
This was a mistake in my reporting, because I did not run the simulations myself. I have corrected it in the text after confirming with the coauthor and checking the original paper.

**page 8, line 204/205:** *This is another - more common - mistake. Although the Mach number is relatively low, a look on the pressure peaks of this case unveils that the slat suction peak (although not shown here but reported in AGARD AR 303 or AGARD*

*CP 515) gets into sonic speed conditions! Therefore, the choice of an incompressible solver for this airfoil is more than questionable.*
Indeed, the choice of an incompressible solver for this benchmark is not entirely proper. Nevertheless, this is only a validation case and a satisfactory match between experiment and numerical predictions is obtained.

*page 8, line 208: The over prediction of the stall angle by $6°$ seems pretty large as the main motivation of the work is based on the prediction of the stall delay by a slat which is mainly the shift in stall angle.*
This is a well known shortcoming of RANS turbulence modeling, but at this point higher-fidelity simulations are too expensive to be used for design cases. Nevertheless, we assume that at least the tendencies - so the sensitivity of lift and drag to changes in the profile shape - are somewhat captured and this is what is important for design optimization.

*page 9, line 214: The conclusion that MSES can be used as a substitute for RANS CFD is weak and not supported. MSES is not able to capture confluent boundary layers at all. Due to the small gap and since the optimum slat position is very close to the position where the confluent boundary layer gets dominant (see Woodward and Lean, AGARD CP515, 1993) an optimization procedure neglecting this effect is likely to predict gaps that are too small.*
The claim that MES is a substitute is based on empirical observations made here, and is not generalizable to other designs more typical for Aerospace applications, in particular with respect to gap width. We have weaked the statement in the main text a bit. Furthermore, the gap width of the obtained designs tended to converge towards the upper bounds, hence confluent boundary layers are not a concern here (even though MSES can not model them). Plus, we use CFD which can predict confluent boundary layers for the performance assessment post-optimization. So if the optimal

gap width obtained from the optimization using MSES was to small, the CFD analysis would make that clear.

**page 10, line 223:** *It is fully unclear why the most sensitive parameter for slat design - the gap - is fixed at the beginning. Additionally, the chosen values seem large. According to Woodward and Lean (1993) an optimum gap is strongly depending on the slat angle and can go down to 2-2.5% chord length. In the further (line 230 and following) the reason for the change in performance is most likely more related to the slat angle than the gap. It is consitent, that the optimal slat deflection angle is lower for the higher gap. At least concerning lift, it does not seem that a maximum lift coeffcient is clearly detected.*
Initially, for the preliminary assessment, we also tried to fix the chord length and leave the gap width variable. However, this just resulted in the gap width converging to the upper bound of the gap width. Then, for the actual design cases, the gap width was initially left variable, but the optimal gap width tended to converge to upper bounds as well. Hence, at some point in order to save on computation time it was just left fixed. But we agree, that indeed the gap width chosen for the preliminary optimization are large. Nevertheless, for the actual designs a gap widths of 2 and 4% were used, respectively.

**page 10, figure 6:** *It is not consistent (and not expected by the reader) to show MSES results in these diagrams. Above it was mentioned, that the designs were optimized by MSES but the performance prediction for the evaluation is done with RANS. Especially, there is no clear max. lift coefficient prediction in the shown data.*
Since this is only a preliminary assessment neither maximum lift nor RANS results are presented.

**page 11, line 258ff:** *an important description needed to understand the figures and*

*the conclusions should not be placed in an appendix.*
The figure was moved into the main text.

*page 12, figure 8: This figure is a collection of all optimization data. It is not very explanative as it overlays too much information. It contains already data (of the integral design) that hasn't yet even been introduced and is decribed much later. This figure should be divided for the different design methods and commented accordingly in the text.*
The figure contains a lot of information such that comparison between the different configurations and the different design procedures is possible. Hence, we propose to leave the figure as is. But we have added a note in the main text to clarify that some of the information in the figure will be discussed later on.

*page 12, lines 263-265: This would be a good option to highlight a common result with previous work (see General comments). This result is also in line with the results obtained by Manso Jaume and Wild for the superimposed slat optimization.*
This is a good remark. An additional section was added (already printed in the general remarks section).

*page 12, lines 263-265: The statement on the sensitivity of separation on the slat shape is not supported by theory. In constast, a cambered plate is less likely to separate at high angles of attack than a flat plate. Additionally, closing the gap increases the slotted airfoil effects in both directions. In fact, as the slat is moved veritcally, the Slat Effect and the Circulation Effect are expected to get stronger. Only the Dumping effect is expected to be reduced due to the reduction of the suction peak due to the strengthened Slat Effect. In consequence, the slat load is increased (higher circulation and higher trailing edge pressure) resulting in a more cambered airfoil to be more suitable to achieve the circulation without separaton. To verify this, a comparison*

*of the pressure distributions is needed. This conlclusion has therefore to be reworked.*
The statement was reworked on (line 287ff, page 13): "Aerodynamic theory indicates that reducing the gap width while avoiding confluent boundary layers leads to an increase in the coupling between the slat and the main element: the slat and the circulation effect are expected to get stronger whereas the dumping effect may be a bit weakened. However, the optimized slat for the lower gap width is less aggressive and the configuration produces less lift, has lower glide ratios and stalls roughly at the same angle of attack. [Nevertheless, this agrees with the literature...]". Indeed, the designs resulting from the reduced gap width are less aggressive which is contrast to the expected increase in the positive coupling between the slat and the main element. Some remarks about this have also been added in the conclusion. It may be interesting to do further investigation into this in a follow-up study.

**page 13, figure 9:** *The line legend of figure 9 introduces an undescribed configuration A\* in addition (same for fig. 14, and in figs 15-17 configs B\*, C\*, D\*). The meaning and origin is perfectly unclear. It can only be assumed from later reading that this configuration related to the integral design that is described much later (starting from page 15). The pressure distributions shown in the right hand side are not discussed at all in the text.*
The legend text has been adjusted to include also the A\* configuration.

**page 14, line 298:** *To be precise, none of the airofils is optimized for maximum lift coefficient. The airfoil optimization only targeted a high lift coeffciient at a high angle of attack (here AoA=$20°$ ). An airfoil stalling at $19°$ could have a higher max- imum lift coefficient than one not stalled at $20°$ . To do a maximum lift coefficient optimization it is necessary to detect the naximum lift coefficient of an airfoil by varying the angle of attack.*
The wording has been adjusted to say maximum lift at the design angles of attack.

*page 14, line 310: Here it is stated that experimental data for the clean airfoil would be available for comparison. Such a comparison would have been an asset in section 3.1 regarding the validation of the methodology.*
We already present two validation cases for multi-element airfoils, we consider this to be sufficient for the publication. Otherwise, the length of the paper is excessively increased.

*page 15, line 318 it should be highlighted that - in contrast to the integrated design work of Manso Jaume and Wild, where the suction side contour of the slat is the contour of the original main airfoil - the integral design here is not restricted by the clean airfoil shapein the same way. This underlined the originality of the present work and its relation to previous work.*
The following adjustment was made in the introduction section: Second, the results of an integral design procedure are shown for thick main elements up to 50 % using a variable spline discretization for both the slat and the main element contrary to the simpler parametrization used by [citation].

*page 17, lines 334/335: It is mentioned that the main airfoil shape is a consequence of the structural constraints. But it is more expected that this is an implicit result of the main airfoil shape optimization. Due to the higher curvature, the suction peak is more locally concentrated (improving the Dumping Effect and stabilizing the slat flow) and the trailing edge position therefore placed close to the maximum curvature - which is now much furhter upstream. The intragrated design by Manso Jaume and Wild shows a similar main airfoil shape, and there, no structural constraint has been imposed.*
While the structural constraints did not have much of an influence on the leading edge and the suction side, the pressure side looked very different without the structural constraints. Namely, the maximum thickness was very far forward followed by a steep

decrease in thickness. Hence, we added this remark to the main text. The comparison to Manso Jaume and Wild with respect to the shape of the main element was also added in the main text.

*page 17, lines 358-360: The discussion describes a "larger leading edge radius". It would be better to descirbe the leading edge as "more blunt". Further on, it is not a larger leading edge radius that shifts the suction peak to the front. A larger leading edge radius alone would reduce the suction peak but not move it. The present description is misleading as the curvature which is responsible for the interaction with the slat is higher (and the radius smaller) and imposes a reduced pressure at the slat trailing edge.*

Indeed, a larger leading edge radius for the same maximum thickness (location) would have been more accurate. We replaced it to say more blunt.

*page 17, line 364/365: It is mentioned that the design angle does not account for lower side separation. This is correct, but anyhow, no clear indication of a lower side separation is seen for the designs with slat even at lower angles of attack.*

The reasoning there was incorrect. We removed the sentence. Since in fact, the shape of the main airfoil obtained from the integral design procedure are less likely to separate on the pressure side than the aft-loaded reference wind turbine airfoils.

*page 17, lines 367/369: This is a very late explanation for a figure that had been placed on page 12. It is necessary to split-up fig. 8 and to place the related illustrations close to the dicsussion in the text.*

Again, splitting up the figure makes it more difficult to compare the trends. But as already mentioned we have added some clarification in the main text when the figure first appears.

***page 17, lines 371:*** *As already discussed "more rounded" suggests a smoother curvature distribution, while the opposite is the case for the integral designs.*
This has been corrected to more blunt.

***page 18, lines 374/375 (more likely lines 369/370):*** *This conclusion is in contrast to a previous statement, where it was concluded that the optimal shape of the slat is not as sensitive to the main airfoil optimization and by this not as affected of the auxilary or integral design method (page 17, lines 356/357).*
Well, we said the shape of slat is not very sensitive to the main airfoil, the location and to some extent also the orientation is. But we have clarified the wording on (page 18, line 375-377): "First, the shape of the slat element excluding angle and position is not very dependent on the shape of the main element, but highly dependent on the optimization boundary conditions.".

***page 18, lines 383-385:*** *Basically, to show the stalling behavior it is necessary to look at the pressure distribution just at stall onset. Best is a comparison with a very low AoA step before and after maximum lift. The used step of $8°$ is too large and - depending on the stall onset angle - the stall is developed over the entire configuration but not showing the onset (main wing or slat or both at the same time).*
That is why the flow fields are shown in the appendix. However, I added two plots with the pressure and skin friction coefficients in the main text. The following answers will also clarify a bit further.

***page 18, line 387:*** *It should be discussed, whether the reattachment due to wake displacement is in accorcance with Snith's 4th effect (Off-Surface Pressure Recovery)*
I do not understand your argumentation here, please explain further.

**page 18, lines 387-389:** *As the stall onset is of primary interest, the discussion of the flow fields should not be placed into an appendix.*
I left the flow fields in the appendix, but I added two plots with pressure and skin friction coefficients before and after maximum lift in the main text.

**page 19, line 393:** *At least here at the end of all shown pressure distributions, it is necessary to conclude about the suitability of the incompressible solver. Although the flow speed is not mentioned (missing in the case description in Table A.1) the level of the pressure coefficient is less than -15 and imposes the need to check whether this assumption is still valid.*
I calculated the free stream Mach number for the NREL 5MW turbine up to about 40 % span and it is below 0.1 which would result in a Glauert correction factor of $c_P/c_{P0} \approx 1.005$. Hence, we did not consider compressible effects, since this is also usually not done for wind energy applications. But I also reran some of the cases with not just accurate Reynolds number, but also accurate Mach number scaling using again the incompressible solver. Then, I checked the maximum Mach number as predicted by the incompressible solver and indeed for the clean cases at angles of attack with lift coefficients above 4, the local Mach number at the slat suction peak approach $0.45$, which is not ideal. However, realistically speaking given that RANS overpredicts the stall angle, I don't think that these conditions will actually be reached. Nevertheless, I added this paragraph before the conclusion: "Some of the designs show very high suction peak values for the slat which is an indication that locally compressibility effects may not be negligible despite the low freestream Mach number of $Ma_{\approx}0.1$. Calculating the local Mach number from the incompressible flow field for all the CFD cases shows that for the designs and angle of attack configuration where the lift coefficient is higher than 4 the Mach number locally approach $0.45$. This is indeed very high and it is recommended that in future publications, compressibility effects should be considered if a design optimization for high lift is carried out. Nevertheless, given that turbulence model of the CFD solver is expected to overpredict the stall

angle, it is not certain that such high Mach number will actually be reached in real life." I also added the freestream Mach number to the table with the design parameters.

***page 19, lines 395-397:*** *The description reverses cause and effect. The high suction peak on the main airfoil is the reason for the low trailing edge pressure at the slat - remind Smith's effects.*
The sentence was reversed.

***page 21, line 428:*** *To conclude on the importance of the gap it should have been used as a design parameter. The limited information on the gap variation (two values only) doesn't allowto draw such abbreviation general conclusion. From other airfoils in literature it is known that the gap is even the most sensitive parameter.*
The sentence was modified to: "A reduction in the gap width did not offer any benefits, but only two gap widths were investigated. Possibly, the sensitivity to this parameter warrants further investigation." But as already mentioned, in initial investigation whenever the gap width was left variable the optimal design would converge to the upper bound even when using bounds up to 10 %. Hence, at some point it was fixed to keep structural loading in check. Also, when looking at the references from Manso Jaume, Schramm, Zahle/Gaunaa (also unpublished work), Pechlivanoglou and Schramm gap widths of the same order of magnitude were used (or obtained from optimization).

**Reply to technical comments**
All the remarks were implemented. With the exception of multi-figure captions as the draft template explicitly asks to remove them. But then later on add them in the full publication. Then the Subsection formation is according to the template guidelines.

---

## Referee Comment (RC2) · Anonymous Referee #2 · 17 Mar 2020

The paper deals with the slat design for thick base profiles at high Reynolds numbers. Due to various combinations of presented cases it is hard to follow the intended logic in the structure. There are too many different cases which are compared back and forth with references to the appendices. The authors should try to better structure the cases and results. After reading the paper I have a hard time to really summarise it for me with a take home message.

Here are a few points that should be addressed.

Page 5. In the shape parametrisation it is written that the leading edge location was fixed to the coordinate system (0,0). Since slat and base profile combined are subject to optimisation it is not clear which leading edge is fixed to (0,0)

[Figure]

Page 6: The optimisation objectives are formulated as a weighted sum of the performance under clean and rough conditions. What do the authors mean by rough conditions and how do they define it?

Page 6: In eq. 5 and 6 the weighting terms have the index "clean / tripped". So far it is not clear what that means. The authors should give more details on the tripping they applied in their simulations. Also, is only the main profile tripped or also the slat?

Page 8/9: The authors try to validate their fluid models against different benchmark cases. In the first one they use MSES and CFD and in the second one they use only CFD. They argue that they can use the lower fidelity model for their optimisation procedure — this is only based on the first benchmark. On the contrary, the authors say that the simple model has problems to converge due to the sharp edges in the geometry. This is a limiting factor in their procedure. So why do the authors also show the second benchmark that does not contribute to their decision?

Page 10: In figure 6 the results for clean and rough are plotted. Again, it is not clear what "rough" refers to and how it is defined.

Page 11: The caption of figure is insufficient, what are e.g. the different lines (dotted, dashed and solid) of the slat?

Page 12: Figure 8: What is L_max, Interm and G_max? What is the "integral design"?

Page 13: Third bullet point: Where is the influence of the base profile thickness discussed?

Page 13: Figure 9: Again, in the caption is stated "rough" and "tripped" without any further description.

Page 14: Second sentence: What do the authors refer to by stating " Hence, the profiles optimized for maximum lift actually perform worse in terms of maximum lift as compared to the ones optimized for maximum glide ratio" ? Where can this be seen in figure 9? Which is the design for lift optimisation and which is the one for maximum

glide ratio?

Page 15: Figure 11: insufficient caption.

Page 17. figure 13: What are the differences on the plots? Even the text doesn't help.

Typos:

page 4, line 94: The second sentence should start with "They" instead of The

page 4, line104: The second sentence at the end of this line should be "Manso Jaume" — there should be no "and" since it is a double name of the author.

---

## Author Comment (AC2) · 19 Mar 2020

**General remarks**

Dear referee,
we appreciate that you took the time to read the paper and give useful feedback that helped us improve the paper. Replies to your remarks follow.

*The paper deals with the slat design for thick base profiles at high Reynolds numbers. Due to various combinations of presented cases it is hard to follow the intended logic in the structure. There are too many different cases which are compared back and forth*

[Figure]

*with references to the appendices. The authors should try to better structure the cases and results. After reading the paper I have a hard time to really summarise it for me with a take home message.*

Clarifications in the text have been added to make it more clear. Additionally, some figures and tables have been moved from the appendix into the main text to make easier to understand.

**Reply to specific comments**
***Page 5:*** *In the shape parametrisation it is written that the leading edge location was fixed to the coordinate system (0,0). Since slat and base profile combined are subject to optimisation it is not clear which leading edge is fixed to (0,0).*
The leading edge of the main element was meant. The text has been modified to clarify this.

***Page 6:*** *The optimisation objectives are formulated as a weighted sum of the performance under clean and rough conditions. What do the authors mean by rough conditions and how do they define it?*
Clean conditions are the conditions where the flow naturally transitions to a turbulence boundary layer. We specify a specific turbulence intensity in the inflow in case of CFD or a specific ampliciation factor in case of MSES which then relates to the transition location. For rough conditions, there is some difference between the results from CFD and MSES. In Mses we specify a specific position on both slat and main element where a transition to a turbulence boundary layer is forced. In CFD, the boundary is assumed to be fully turbulent so there is no transition. Clarification was added to text on page 6.

***Page 6:*** *In eq. 5 and 6 the weighting terms have the index "clean / tripped". So far it is not clear what that means. The authors should give more details on the tripping they*

*applied in their simulations. Also, is only the main profile tripped or also the slat?*
Clarification was added in the text on pages 6 and 7. Both main and slat element are tripped.

*Page 8/9: The authors try to validate their fluid models against different benchmark cases. In the first one they use MSES and CFD and in the second one they use only CFD. They argue that they can use the lower fidelity model for their optimisation procedure as this is only based on the first benchmark. On the contrary, the authors say that the simple model has problems to converge due to the sharp edges in the geometry. This is a limiting factor in their procedure. So why do the authors also show the second benchmark that does not contribute to their decision?*
MSES has been used in literature on cases with sharp edges, but we could not make it work. That is why it was left out. The second benchmark case is used to show that the CFD model can yield accurate drag predictions, because there are issues with this for the first benchmark case. This helps validate our hypothesis that there are issue with the drag measurements for the first benchmark case.

*Page 10: In figure 6 the results for clean and rough are plotted. Again, it is not clear what "rough" refers to and how it is defined.*
This should be clear now, since clarification was added on pages 6 and 7. Also see the previous replies.

*Page 11, Figure 7: The caption of the Figure is insufficient, what are e.g. the different lines (dotted,dashed and solid) of the slat?*
The legend has been extended.

*Page 12, Figure 8: What is Lmax, Interm and Gmax? What is the "integral design"?*

The integral designs are obtained in an optimization procedure where both the slat and the main element are optimized simulataneously. A remark has been added in the text to say that the results of the integral design procedure are not relevant yet at this point in the text. Lmax, Interm and Gmax refer to the location of the designs on the pareto front as shown in the previous. This figure has been moved from the appendix to the main text, to make it more clear.

*Page 13: Third bullet point: Where is the influence of the base profile thickness discussed?*
It says in the text with respect to the slat design, "The change in the base profile thickness introduces smaller design deviations from the baseline case as compared to the chord and gap width reduction. This goes back to the argument that the strongest design driver for the slat element is the location of the suction peak on the main element."

*Page 13: Figure 9: Again, in the caption is stated "rough" and "tripped" without any further description.*
This has been explained in previous replies.

*Page 14: Second sentence: What do the authors refer to by stating " Hence, the profiles optimized for maximum lift actually perform worse in terms of maximum lift as compared to the ones optimized for maximum glide ratio" ? Where can this be seen in figure 9? Which is the design for lift optimisation and which is the one for maximum glide ratio?*
The legend of Figure 9 has been extended to clarfiy a bit more. What the figure shows is that MSES overpredicts the stall angle and hence the maximum lift as compared to CFD for the cases where we optimized for maximum lift. For the profiles optimized for maximum glide ratio, MSES and CFD predict similar stall angles. Hence, these

designs actually yield higher lift.

*Page 15, Figure 11 :* *insufficient caption.*
The caption has been modified.

*Page 17, Figure 13:* *What are the differences on the plots? Even the text doesn't help.*
The figure shows the designs resulting from the auxiliary and integral design procedure given the different optimization boundary conditions.

**Reply to technical comments**
There were only minor remarks and they have been corrected.

---

## Referee Report (RR1)

**General comments**

The work described in the paper is a very interesting design approach. The work as such is done in a good scientific manner. The wording is clear and good to read and understand. The manuscript is clearly improved by the revisions made by the authors.

**replies to author's response & specific comments to revision**

- page 2, line 51 *The description of the Dumping Effect does not describe the origin of the accelerated flow. It must be described that the high velocity at the forward element trailing edge is induced by the low pressure of the suction region at the leading edge of the downstream element.*

  This was reformulated to: "This effect is closely related to the circulation effect. The circulation around the main element also leads to a low pressure region around the slat trailing edge. As a consequence the high outflow velocity of the boundary layer of the slat relieves the adverse pressure gradient on the slat element. Hence, separation problems are further alleviated."

  *The description of the Dumping Effect still needs a slight improvement to be fully correct: "The circulation around the main element induces a low pressure region around the main element leading edge, and thus at the slat trailing edge located in its vicinity.*

- page 7, line 191: *It is stated that O-mesh topologies are applied although Pointwise is used. Please state, why not a C-mesh is used that would allow an improved capturing of the slat and main airfoil wakes.*

  This is a good remark and should be considered in further publications. However, in this paricular case with the automated meshing procedure using an O-mesh was more practical. Further, a thorough mesh sensitivity study was carried out.

  *performing a mesh sensitivity study will not unviel the specific difference of the meshing approach. An O-mesh will never be able to resolve the wake over a long distance. It's o.k. for the present work but keep this in mind for future studies.*

- page 8, line 204/205: *This is another - more common - mistake. Although the Mach number is relatively low, a look on the pressure peaks of this case unveils that the slat suction peak (although not shown here but reported in AGARD AR 303 or AGARD CP 515) gets into sonic speed conditions! Therefore, the choice of an incompressible solver for this airfoil is more than questionable.*

  Indeed, the choice of an incompressible solver for this benchmark is not entirely proper. Nevertheless, this is only a validation case and a satisfactory match 4between experiment and numerical predictions is obtained.

  *It is still recommended to delete the sentence refering to Sorensen, which is now on page 9 in line 213/214. Or replace the two sentences by something*

*like* " Although freestream Mach number is low, comperssibility of the air
flow may effect the slat flow due to the high acceleration of the flow. Nev-
ertheless, as will be seen later on, good agreement between experimental
and numerical results is obtained by an incompressible solver."

- page 8, line 208: *The over prediction of the stall angle by 6∘ seems pretty
  large as the main motivation of the work is based on the prediction of the
  stall delay by a slat which is mainly the shift in stall angle.*

  This is a well known shortcoming of RANS turbulence modeling, but
  at this point higher-fidelity simulations are too expensive to be used for
  design cases. Nevertheless, we assume that at least the tendencies - so the
  sensitivity of lift and drag to changes in the profile shape - are somewhat
  captured and this is what is important for design optimization.

  *A disagreement of 6° in stall onset prediction cannot be "satisfactory". It
  is still recommended to either replace "satisfactory" by something weaker
  and to delete the "stall" in the sentence. Acceptable can only be the lift
  prediction in the linear incidence range*

- page 9, line 214: *The conclusion that MSES can be used as a substitute
  for RANS CFD is weak and not supported. MSES is not able to cap-
  ture confluent oundary layers at all. Due to the small gap and since the
  optimum slat position is very close to the position where the confluent
  boundary layer gets dominant (see Woodward and Lean, AGARD CP515,
  1993) an optimization procedure neglecting this effect is likely to predict
  gaps that are too small.*

  The claim that MES is a substitute is based on empirical observations
  made here, and is not generalizable to other designs more typical for
  Aerospace applications, in particular with respect to gap width. We have
  weaked the statement in the main text a bit. Furthermore, the gap width
  of the obtained designs tended to converge towards the upper bounds,
  hence confluent boundary layers are not a concern here (even though
  MSES can not model them). Plus, we use CFD which can predict con-
  fluent boundary layers for the performance assessment post-optimization.
  So if the optimal gap width obtained from the optimization using MSES
  was to small, the CFD analysis would make that clear.

  *Still, a "substitute" would provide equal qualitiy results. Instead, MSES
  in this case is expected to provide similar trends (meaning sensitivities not
  values) at less effort and acceptedly less accuracy - in practice a "lower
  order" or "low-fidelity" or "fast prediction" method*

- page 10, line 223: *It is fully unclear why the most sensitive parameter for
  slat design - the gap - is fixed at the beginning. Additionally, the chosen
  values seem large. According to Woodward and Lean (1993) an optimum
  gap is strongly depending on the slat angle and can go down to 2-2.5%
  chord length. In the further (line 230 and following) the reason for the
  change in performance is most likely more related to the slat angle than*

*the gap. It is consitent, that the optimal slat deflection angle is lower for the higher gap. At least concerning lift, it doesn't seem that a maximum lift coeffcient is clearly detected.*

Initially, for the preliminary assessment, we also tried to fix the chord length and leave the gap width variable. However, this just resulted in the gap width converging to the upper bound of the gap width. Then, for the actual design cases, the gap width was initially left variable, but the optimal gap width tended to converge to upper bounds as well. Hence, at some point in order to save on computation time it was just left fixed. But we agree, that indeed the gap width chosen for the preliminary optimization are large. Nevertheless, for the actual designs a gap widths of 2 and 4% were used, respectively.

*In this case, this should be mentioned by a sentence in the text: "initial experience showed that the gap converges to its upper bound". It is better to not give the impression, that the fixation was made by luck.*

- page 18, line 387: *It should be discussed, whether the reattachment due to wake displacement is in accorcance with Smiths 4th effect (Off-Surface Pressure Recovery)*

I do not understand your argumentation here, please explain further.

*I refer to the sentence "At some point, the slat wake extent grows so much that the low-pressure area in the wake leads to reattachment of the flow on the main element." It has been well explained by Smith that the pressure recovery in a free shear layer wake flow is more "efficient" than in a wall attached boundary layer. While running into the pressure rise, the kinetic energy of the weakest area in the wake can be eaten up even before the pressure has completely recovered to infinity static pressure. If the shear layer is wall attached, this leads to separation. If it is a free wake, the flow stops and a stand-still area of the flow is observed. The further pressure rise is then solely obtained by pressure diffusion. In consequence, the resulting dead-water area displaces the flow leading to a lower pressure above the wing. This in consequence relaxes the pressure gradient along the main wing and the flow stays attached. In highly affected cases, this can be observed when plotting the trailing edge pressure over the angle of attack. There you may detect a reduction of pressure once the effect sets in.*

- page 19, line 393: *At least here at the end of all shown pressure distributions, it is necessary to conclude about the suitability of the incompressible solver. Although the flow speed is not mentioned (missing in the case description in Table A.1) the level of the pressure coefficient is less than -15 and imposes the need to check whether this assumption is still valid.*

... However, realistically speaking given that RANS overpredicts the stall angle, I don't think that these conditions will actually be reached.

And this is exactly the point I wanted to make. Don't think that this is out of reality. We observed even transonic flows at slat devices at onflow Mach number of M=0.2. Anyhow, the text addition made in the revision is clear and accepted